# WaGI: Wavelet-based GAN Inversion for Preserving High-Frequency Image Details

## Abstract

Recent GAN inversion models focus on preserving image-specific details through various methods, *e.g.*, generator tuning or feature mixing. While those are helpful for preserving details compared to a naïve low-rate latent inversion, they still fail to maintain high-frequency features precisely. In this paper, we point out that the existing GAN inversion models have inherent limitations in both structural and training aspects, which preclude the delicate reconstruction of high-frequency features. Especially, we prove that the widely-used loss term in GAN inversion, *i.e.*, $L_2$, is biased to reconstruct low-frequency features mainly. To overcome this problem, we propose a novel GAN inversion model, coined WaGI, which enables to handle high-frequency features explicitly, by using a novel wavelet-based loss term and a newly proposed wavelet fusion scheme. To the best of our knowledge, WaGI is the first attempt to interpret GAN inversion in the frequency domain. We demonstrate that WaGI shows outstanding results on both inversion and editing, compared to the existing state-of-the-art GAN inversion models. Especially, WaGI robustly preserves high-frequency features of images even in the editing scenario. We will release our code with the pre-trained model after the review.

## 1 Introduction

Recently, the inversion of Generative Adversarial Networks (GANs) (Goodfellow et al., 2020) has dramatically improved by using the prior knowledge of powerful unconditional generators (Karras et al., 2019; 2020; 2021) for the robust and disentangled image attribute editing (Abdal et al., 2019; Richardson et al., 2021; Tov et al., 2021; Alaluf et al., 2021a;b; Wang et al., 2022a). The early GAN inversion models mostly rely on per-image optimization (Abdal et al., 2019; 2020; Zhu et al., 2020), which is extremely time-consuming. For real-time inference, the encoder-based GAN inversion method becomes prevalent (Richardson et al., 2021; Tov et al., 2021; Alaluf et al., 2021a; Moon & Park, 2022), which trains an encoder that returns the corresponding GAN latent of an input image. The acquired latent from the encoder is desired to reproduce the input image as closely as possible.

However, the encoder needs to compress the image into a small dimension, *i.e.*, low-rate inversion. For instance, in the case of StyleGAN2 (Karras et al., 2020), for encoding an image with the size $1024^2 \times 3$ the encoders return the corresponding latent with the size $18 \times 512$, which is extremely smaller than the original image dimension (about 0.3%). Due to the Information Bottleneck theory (Tishby & Zaslavsky, 2015; Wang et al., 2022a), an attempt to encode information into a small tensor occurs severe information loss, and it deteriorates the image details, *i.e.*, high-frequency features.

To overcome this shortage, recent GAN inversion models propose new directions, such as fine-tuning the generator (Roich et al., 2021; Alaluf et al., 2021b) or directly manipulating the intermediate feature of the generative model (Wang et al., 2022a) to deliver more information using higher dimensional features than latents, *i.e.*, high-rate inversion. However, results of high-rate inversion models are still imperfect. Figure 1 shows the inversion results of high-rate inversion models, HyperStyle (Alaluf et al., 2021b), and HFGI (Wang et al., 2022a). Though both models generally preserve coarse features, the details are distorted, *e.g.*, boundaries of the letter and the shape of accessories.

The aforementioned high-rate inversion models remarkably decrease distortion compared to the state-of-the-art low-rate inversion models, *i.e.*, Restyle (Alaluf et al., 2021a). However, this does not mean that distortion on every frequency spectrum is *evenly* decreased. To explicitly check distortion

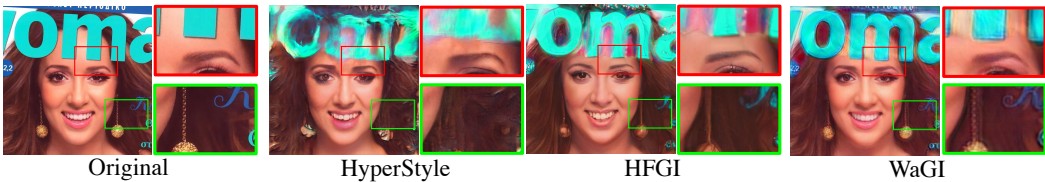

Original           HyperStyle           HFGI           WaGI

Figure 1: **Preserving details at the image inversion.** Comparison of inversion results for the noisy image. The zoomed parts are regions that require delicate preservation of details. The existing GAN inversion models including recent high-rate inversion methods, such as generator tuning, *e.g.*, HyperStyle, and feature mixing, *e.g.*, HFGI, still struggles to restore high-frequency details.

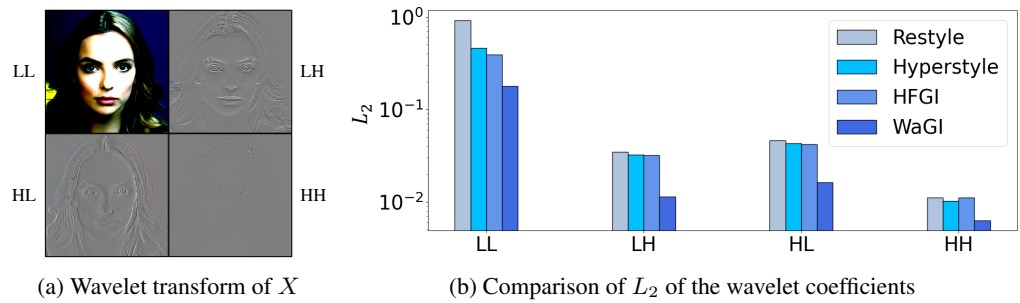

(a) Wavelet transform of $X$           (b) Comparison of $L_2$ of the wavelet coefficients

Figure 2: **Wavelet transform and $L_2$ loss from each filter.** (a) We plot the wavelet coefficients by each filter at $1^{st}$ wavelet decomposition. The gray color denotes the zero value. Coefficients from $LH$, $HL$, and $HH$, have significantly high sparsity than the coefficient from $LL$. (b) We plot the average $L_2$ of each wavelet coefficient between CelebA-HQ test images and corresponding inverted images by various state-of-the-art inversion models. Due to the significant gap between $L_{2,LL}$ and the rest (about 30 times), we display the losses with the logarithmic scale for better visualization.

per frequency sub-bands, we adopt a wavelet transform, which enables to use of both frequency and spatial information. The wavelet transform yields a total of four coefficients by passing filters, which are in a low-pass filter set $\mathbb{F}_l = \{LL\}$ and a high-pass filter set $\mathbb{F}_h = \{LH, HL, HH\}$. In Figure 2a, we visualize the coefficients obtained by each filter. In Figure 2b, we compare $L_2$ between the coefficients of ground truth images and inverted images, yielded by filter $f$, *i.e.*, $L_{2,f}$. While the high-rate inversion models apparently decrease $L_{2,f}$ for $f \in \mathbb{F}_l$, they marginally decrease or even increase $L_{2,f}$ for $f \in \mathbb{F}_h$, compared to Restyle. In the light of this observation, we can argue that the existing methods of increasing the information rate can decrease distortion on the low-frequency sub-band, but are not effective for decreasing distortion on the high-frequency sub-band.

**Contributions** First, we prove that the widely used loss term in GAN inversions, *i.e.*, $L_2$, is biased on low-frequency by using the wavelet transform. Then, we propose a simple wavelet-based GAN inversion model, named WaGI, which effectively lowers distortions on both the low-frequency and high-frequency sub-bands. To the best of our knowledge, WaGI is the first attempt to interpret GAN inversion in the frequency domain. Especially, we propose two novel terms for our model: (i) wavelet loss and (ii) wavelet fusion. First, (i) amplifies the loss of the high-frequency sub-band by using the wavelet coefficients from $f \in \mathbb{F}_h$. By using (i) at training, WaGI is proficient in reconstructing high-frequency details. Second, (ii) transfers the high-frequency features directly to the wavelet coefficients of the reconstructed image. Due to the wavelet upsampling structure of SWAGAN, we can explicitly manipulate the wavelet coefficients during the hierarchical upsampling.

We demonstrate that WaGI shows outstanding results, compared to the existing state-of-the-art GAN inversion models(Alaluf et al., 2021b; Wang et al., 2022a). We achieve the lowest distortion among the existing GAN inversion models on the inversion scenario. Moreover, qualitative results show the robust preservation of image-wise details of our model, both on the inversion and editing scenarios via InterFaceGAN (Shen et al., 2020) and StyleCLIP (Patashnik et al., 2021). Finally, we elaborately show the ablation results and prove that each of our proposed methods is indeed effective.

## 2 RELATED WORK

### 2.1 WAVELET TRANSFORM

Wavelet transform provides information on both frequency and spatiality (Daubechies, 1990), which is crucial in the image domain. The most widely-used wavelet transform in deep learning-based image processing is the Haar wavelet transform, which contains the following simple four filters:

$$LL = \begin{bmatrix} 1 & 1 \\ 1 & 1 \end{bmatrix}, LH = \begin{bmatrix} -1 & -1 \\ 1 & 1 \end{bmatrix}, HL = \begin{bmatrix} -1 & 1 \\ -1 & 1 \end{bmatrix}, HH = \begin{bmatrix} 1 & -1 \\ -1 & 1 \end{bmatrix}. \tag{1}$$

Since the Haar wavelet transform enables reconstruction without information loss via inverse wavelet transform (Yoo et al., 2019), it is widely used in image reconstruction-related tasks, *e.g.*, super-resolution (Huang et al., 2017; Liu et al., 2018) and photo-realistic style transfer (Yoo et al., 2019). In GAN inversion, image-wise details that cannot be generated via unconditional generator, *e.g.*, StyleGAN (Karras et al., 2019; 2020), should be transferred without information loss. Consequently, we argue that the Haar wavelet transform is appropriate for GAN inversion. To the best of our knowledge, our method is the first approach to combine the wavelet transform in GAN inversion.

### 2.2 FREQUENCY BIAS OF GENERATIVE MODELS

Recent works (Dzanic et al., 2019; Liu et al., 2020; Wang et al., 2020) tackle the spectral bias of GANs, in that GAN training is biased to learn the low-frequency distribution whilst struggling to learn the high-frequency counterpart. GANs suffer from generating high-frequency features due to incompatible upsampling operations in the pixel domain (Schwarz et al., 2021; Frank et al., 2020; Khayatkhoei & Elgammal, 2022). In order to alleviate the spectral distortions, prior works (Schwarz et al., 2021; Durall et al., 2020; Jiang et al., 2021) propose a spectral regularization loss term to match the high-frequency distribution. However, we empirically find that images generated via spectral loss can contain undesirable high-frequency noises to coercively match the spectrum density, eventually degrading the image quality (see Appendix A.2.1).

Currently, SWAGAN (Gal et al., 2021) is the first and only StyleGAN2-based architecture that generates images directly in the frequency domain to create fine-grained contents in the high-frequency range. The hierarchical growth of predicted wavelet coefficients contributes to the accurate approximation of spectral density, especially in the spatial domain for realistic visual quality. We adopt this wavelet-based architecture to preserve high-frequency information and mitigate spectral bias.

### 2.3 HIGH-RATE GAN INVERSION

The most popular high-rate GAN inversion methods are generator tuning and intermediate feature fusion. First, generator tuning is first proposed by Pivotal Tuning (Roich et al., 2021), which fine-tunes the generator to lower distortion for input images. Since Pivotal Tuning needs extra training for every new input, it is extremely time-consuming. Recently, HyperStyle (Alaluf et al., 2021b) enables generator tuning without training, by using HyperNetwork (Ha et al., 2016). Though HyperStyle effectively lowers distortion compared to low-rate inversion methods, it still encounters frequency bias. Second, intermediate feature fusion is proposed by HFGI (Wang et al., 2022a). HFGI calculates the missing information of low-rate inversion and extracts feature vectors via a consultant encoder. The extracted feature is fused with the original StyleGAN feature, to reflect image-specific details into the decoding process. However, HFGI relies on the low-frequency biased loss term even when training the image-specific delivery module, which leads to frequency bias.

## 3 METHOD

In this section, we propose an effective Wavelet-based GAN Inversion method, named WaGI. In Section 3.1, we first introduce the notation and the architecture of WaGI. In Section 3.2, we prove the low-frequency bias in $L_2$, using the Haar wavelet transform. In addition, to alleviate frequency bias, we propose a novel loss term, named *wavelet loss*. Lastly, we point out the limitation of existing feature fusion and propose *wavelet fusion*, which robustly transfers high-frequency features.

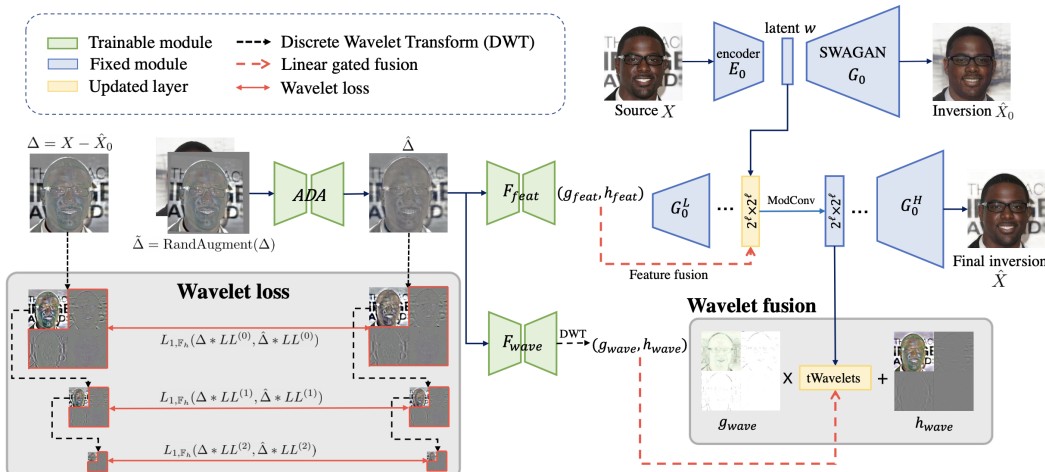

Figure 3: **Training scheme of Wavelet-based GAN Inversion (WaGI).** Given a pre-trained encoder $E_0$ and generator $G_0$, we can obtain an initial inverted image $\hat{X}_0$. The residual $\Delta$ contains high-fidelity details that $\hat{X}_0$ misses. The model leverages a trainable $ADA$ module to align the residual, which should ultimately be in alignment with $\hat{X}_0$ or the edited image $\hat{X}_0^{edit}$ at inference. From the aligned $\hat{\Delta}$, we can replenish the missing high-fidelity information with the two fusion modules $F_{feat}$ and $F_{wave}$. Fusion with each output is operated in the feature and spatial frequency domains in separate intermediate layers between lower layers $G_0^L$ and higher layers $G_0^H$ of $G_0$, respectively. The final inversion result $\hat{X}$ contains rich information without the loss of high-frequency components. Note that $ADA$, $F_{feat}$, and $F_{wave}$ are all jointly trained, while $E_0$ and $G_0$ are frozen.

## 3.1 NOTATION AND ARCHITECTURE

Figure 3 shows our overall architecture. Since the goal of WaGI is to retain high-frequency details of an image $X$, we design our model to be a two-stage; naïve inversion and image-wise detail addition.

First, the pre-trained SWAGAN generator $G_0$ and its pre-trained encoder $E_0$[1] can obtain low-rate latent, $w = E_0(X)$, and the corresponding naïve inversion $\hat{X}_0 = G_0(w)$. Due to the inherent limitation of low-rate inversion, $\hat{X}_0$ misses image-wise details, denoted by $\Delta = X - \hat{X}_0$. Since the edited image, $\hat{X}_0^{edit}$, might have distorted alignment, *e.g.*, a varied face angle, eye location, etc., $\Delta$ should adapt to the alignment of $\hat{X}_0^{edit}$ at the editing scenario. Hence, we train the Adaptive Distortion Alignment (Wang et al., 2022a) ($ADA$) module that re-aligns $\Delta$ to fit the alignment of $\hat{X}_0^{edit}$. At the training, due to the absence of edited images preserving image-wise details, we impose a self-supervised task by deliberately making a misalignment via a random distorted map to $\Delta$, $\tilde{\Delta} = \text{RandAugment}(\Delta)$ (Wang et al., 2022a). The purpose of $ADA$ is to minimize the discrepancy between output $\hat{\Delta} = ADA(\hat{X}_0, \tilde{\Delta})$ and $\Delta$.

WaGI combines $\hat{\Delta}$ using the low-rate latent $w$ with feature fusion, $F_{feat}$ (Wang et al., 2022a), and our proposed method, wavelet fusion, $F_{wave}$. Both $F_{feat}$ and $F_{wave}$ follow the linear gated scheme to filter out redundant information, which return the pairs (*scale, shift*), *i.e.*, $(g_{feat}, h_{feat}) = F_{feat}(\hat{\Delta})$ and $(g_{wave}, h_{wave}) = F_{wave}(\hat{\Delta})$, respectively. The pairs of $g$ and $h$ adaptively merge desired information from $w$ and $\hat{\Delta}$. Refer to Section 3.3 for the detailed processes of fusions.

The overall loss merges self-supervised alignment loss $L_{ADA}$, wavelet loss $L_{wave}^K$ between $X$ and the final image $\hat{X}$, and reconstruction loss $L_{image}(\lambda_{L_2}, \lambda_{id}, \lambda_{LPIPS})$, consisting of the weighted sum of $L_2$, LPIPS (Zhang et al., 2018), and $L_{id}$, with weights $\lambda_{L_2}$, $\lambda_{id}$, and $\lambda_{LPIPS}$, respectively:

$$\mathcal{L}_{total} = L_{ADA}(\Delta, \hat{\Delta}) + L_{Image}(\lambda_{L_2}, \lambda_{id}, \lambda_{LPIPS})(X, \hat{X}) + \lambda_{wave} L_{wave}^K(X, \hat{X}). \quad (2)$$

---

[1]We used $e4e$(Tov et al., 2021) for $E_0$.

## 3.2 WAVELET LOSS

Majority of GAN inversion models use $L_{Image}(\lambda_{L_2}, \lambda_{id}, \lambda_{LPIPS})$ with weights $\lambda_{L_2} > \lambda_p > \lambda_{id}$, for comparing the ground truth and the reconstructed image (Richardson et al., 2021; Tov et al., 2021; Alaluf et al., 2021a; Roich et al., 2021; Alaluf et al., 2021b; Wang et al., 2022a). Combining the following theorem and our observation, we demonstrate that $L_2$, which has the highest weight in $L_{Image}$, is biased on the low-frequency sub-band in terms of the Haar wavelet transform:

**Theorem 1.** *The following equation holds when $\lambda_f = 1$, $\forall f \in \mathbb{F}_l \cup \mathbb{F}_h$:*

$$L_2(I_1, I_2) = \sum_{f \in \mathbb{F}_l \cup \mathbb{F}_h} \lambda_f L_{2,f}(I_1, I_2), \tag{3}$$

*where $I_1$ and $I_2$ are arbitrary image tensors. Proof is in Appendix A.1.*

We can derive a following simple lemma using Theorem 1.

**Lemma 1.1.** *When the distributions of pixel-wise differences between $I_1$ and $I_2$ are i.i.d., and follow $\mathcal{N}(\mu, \sigma^2)$ with $\mu \approx 0$, the following equation holds when $\lambda_f = \frac{1}{4}$, $\forall f \in \mathbb{F}_l \cup \mathbb{F}_h$:*

$$\log \mathbb{E}[L_1(I_1, I_2)] + C = \sum_{f \in \mathbb{F}_l \cup \mathbb{F}_h} \lambda_f \log \mathbb{E}[L_{1,f}(I_1, I_2)], \tag{4}$$

*where $C$ is a constant. Proof is in Appendix A.1.*

In Theorem 1, since $L_2$ reflects $L_{2, f \in \mathbb{F}_l \cup \mathbb{F}_h}$ with the same weight, it seems a fair loss without frequency bias. However, as shown in Figure 2b, we empirically find that $L_{2,LL}$ of the existing GAN inversion models, *e.g.*, Restyle (Alaluf et al., 2021a), HyperStyle (Alaluf et al., 2021b), and HFGI (Wang et al., 2022a), have around 30 times larger value than $L_{2,f \in \mathbb{F}_h}$, on average. The same logic can be applied to $L_1$, which shows the extreme low-frequency bias while using the same $\lambda_f$ for the equation, as shown in Lemma 1.1 (Please refer to Appendix A.1 for the detailed descriptions). Consequently, we can argue that $\lambda_{f \in \mathbb{F}_h}$ should be higher than $\lambda_{LL}$ to avoid the low-frequency bias.

In the light of this observation, we design a novel loss term, named wavelet loss, to focus on the high-frequency details. The wavelet loss $L_{wave}$ between $I_1$ and $I_2$ is defined as below:

$$L_{wave}(I_1, I_2) = \sum_{f \in \mathbb{F}_h} L_{2,f}(I_1, I_2). \tag{5}$$

$I_1$ and $I_2$ in equation 5 only pass $f \in \mathbb{F}_h$, which discards the sub-bands with the frequency below $f_{nyq}/2$, where $f_{nyq}$ is the Nyquist frequency of the image. However, we empirically find that a substantial amount of image details are placed below $f_{nyq}/2$. In other words, $L_{wave}$ should cover a broader range of frequency bands. To this end, we improve $L_{wave}$ combining with multi-level wavelet decomposition (Liu et al., 2018; Yoo et al., 2019), which can subdivide the frequency ranges by iteratively applying four filters to the $LL$ sub-bands. We improve $L_{wave}$ with the $K$-level wavelet decomposition, named $L_{wave}^K$:

$$L_{wave}^K(I_1, I_2) = \sum_{i=0}^{K} \sum_{f \in \mathbb{F}_h} L_{2,f}((I_1 * LL^{(i)}), (I_2 * LL^{(i)})), \tag{6}$$

where $LL^{(i)}$ stands for iteratively applying $LL$ for $i$ times for multi-level wavelet decomposition. $L_{wave}^K$ can cover the image sub-bands with the frequency ranges between $f_{nyq}/2^{K+1}$ and $f_{nyq}$.

Now we propose that $L_{wave}^K$ is especially helpful for training modules that address high-frequency details, *e.g.*, $ADA$. The existing $ADA$ in HFGI uses $L_1(\tilde{\Delta}, \Delta)$, but we find that this loss term is not enough to focus on high-frequency features. Since $ADA$ targets to re-align high-frequency details to fit the alignment of the edited image, high-frequency details should be regarded for both purposes; fitting alignment and preserving details. Consequently, we add the wavelet loss to train $ADA$:

$$L_{ADA}(I_1, I_2) = L_1(I_1, I_2) + \lambda_{wave,ADA} L_{wave}^K(I_1, I_2). \tag{7}$$

We set $\lambda_{wave,ADA} = 0.1$ and $K = 2$ for training.

### 3.3 WAVELET FUSION

To prevent the generator from relying on the low-rate latent, *i.e.*, $w \in W+$, we should delicately transfer information from $\tilde{\Delta}$ to the generator. For instance, HFGI (Wang et al., 2022a) extracts *scale* $g_{feat}$ and *shift* $h_{feat}$ using $\tilde{\Delta}$, and fuse them with $F_{\ell_f}$, the original StyleGAN intermediate feature at layer $\ell_f$, and the latent at layer $\ell_f$, $w_{\ell_f}$ as follows:

$$F_{\ell_f+1} = g_{feat} \cdot \text{ModConv}(F_{\ell_f}, w_{\ell_f}) + h_{feat}. \tag{8}$$

Though feature fusion is helpful for preserving the image-specific details, the majority of image details, *e.g.*, exact boundaries, are still collapsed (see Section 4 for more details). We attribute this to the low-resolution of feature fusion. In the case of HFGI, the feature fusion is only applied to the resolution of 64, *i.e.*, $\ell_f = 7$. According to the Shannon-Nyquist theorem, the square image $I \in \mathbb{R}^{H \times W}$, where $H = W = l$, cannot store the information with the frequency range higher than $f_{nyq} = \sqrt{H^2 + W^2} = l\sqrt{2}$. Consequently, the upper bound of information frequency by the feature fusion of HFGI is $f_{nyq} = 64\sqrt{2}$, which is relatively lower than the image size, 1024. To solve this, we modify the feature fusion to be done on both 64 and 128 resolutions, i.e., $\ell_f = 7$ and 9, respectively, which means $f_{nyq}$ is doubled.

However, a simple resolution increment cannot address the problem totally (see Section 4.3 for more details). Since feature fusion goes through the pre-trained convolution of the generator, the degradation of image-specific details is inevitable. To avoid degradation, we propose a novel method, named wavelet fusion. Wavelet fusion directly handles the wavelet coefficients instead of the generator feature.

Using the hierarchical upsampling structure of SWAGAN which explicitly constructs the wavelet coefficients, wavelet fusion can transfer high-frequency knowledge without degradation. Similar with feature fusion, our model obtains *scale* $g_{wave}$ and *shift* $h_{wave}$ using $\tilde{\Delta}$, and fuse them with the wavelet coefficients obtained by the layer, tWavelets as follows:

$$\mathcal{W}_{\ell_w} = g_{wave} \cdot \text{tWavelets}(F_{\ell_w}) + h_{wave}, \tag{9}$$

where $\mathcal{W}_{\ell_w}$ are the wavelet coefficients of the $\ell_w$-th layer.

We empirically find that feature fusion is helpful for reconstructing the coarse shape, while wavelet fusion is helpful for reconstructing the fine-grained details. Consequently, we apply feature fusion for the earlier layers ($\ell_f = 7$ and 9) than wavelet fusion ($\ell_w = 11$).

## 4 EXPERIMENTS

In this section, we compared the results of WaGI with various GAN inversion methods. We used the widely-used low-rate inversion models, *e.g.*, *pSp* (Richardson et al., 2021), *e4e* (Tov et al., 2021), and Restyle (Alaluf et al., 2021a), together with the state-of-the-art high-rate inversion models, *e.g.*, HyperStyle (Alaluf et al., 2021b) and HFGI (Wang et al., 2022a) as baselines. First, we compared the quantitative performance of inversion results. Next, we compared the qualitative results of inversion and editing scenarios. For editing, we used InterFaceGAN (Shen et al., 2020) and StyleCLIP (Patashnik et al., 2021) to manipulate the latents. Finally, we analyzed the effectiveness of our wavelet loss and wavelet fusion. We conducted all the experiments in the human face domain: we used the Flickr-Faces-HQ (FFHQ) dataset (Karras et al., 2019) for training and the CelebA-HQ dataset (Karras et al., 2017) for evaluation, with all images generated to the high-resolution $1024 \times 1024$. We trained our own *e4e* encoder, InterFaceGAN boundaries, and StyleCLIP for each attribute to exploit the latent space of SWAGAN (Gal et al., 2021).

### 4.1 QUANTITATIVE EVALUATION

We first evaluated our inversion qualities with the existing baseline inversion methods. Table 1 shows the quantitative results of each method. We used all 3k images in the test split of the CelebA-HQ dataset and evaluated them with (i) the standard objectives including $L_2$, LPIPS, SSIM, and ID similarity, and (ii) the wavelet loss (equation 6) to measure the spatial frequency distortion in the

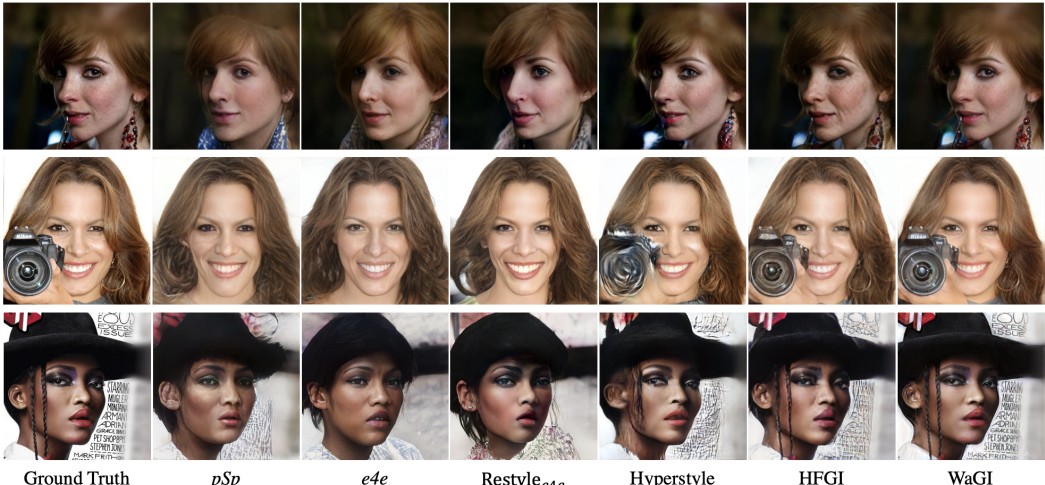

| Ground Truth | *pSp* | *e4e* | Restyle$_{e4e}$ | Hyperstyle | HFGI | WaGI |

Figure 4: **Qualitative comparison between inversion results of baselines.** The baseline models including the state-of-the-art high-rate inversion models failed to robustly preserve details, such as accessories and complex backgrounds. In contrast, the inverted images through WaGI showed robust reconstruction of image-wise details, *e.g.*, legible letters on the second and third rows.

| Method | $L_2 \downarrow$ | $L_{wave} \downarrow$ | LPIPS $\downarrow$ | SSIM $\uparrow$ | ID sim $\uparrow$ |
|---|---|---|---|---|---|
| *pSp* (Richardson et al., 2021) | 0.039 | 0.369 | 0.369 | 0.644 | 0.810 |
| *e4e* (Tov et al., 2021) | 0.053 | 0.390 | 0.419 | 0.591 | 0.786 |
| Restyle$_{e4e}$ (Alaluf et al., 2021a) | 0.049 | 0.389 | 0.383 | 0.638 | 0.783 |
| HyperStyle (Alaluf et al., 2021b) | 0.027 | 0.354 | 0.332 | 0.652 | 0.831 |
| HFGI (Wang et al., 2022b) | 0.023 | 0.351 | 0.323 | 0.661 | 0.864 |
| **WaGI** | **0.011** | **0.230** | **0.277** | **0.753** | **0.906** |

Table 1: **Quantitative comparison between inversion results of baselines.** WaGI outperformed all of the baselines consistently with a large margin in the various metrics, related to both reconstruction quality and fidelity. Quantitative results showed that our model learned the ground truth frequency distribution most accurately, without loss of identity and perceptual quality.

high-frequency sub-band. Our model consistently outperformed all of the baselines with a large margin. The results empirically prove that our wavelet loss and wavelet fusion in the spatial frequency domain are capable of minimizing the spectral distortion and thus improving the reconstruction quality. We note, interestingly, that our model recorded the lowest LPIPS loss despite the explicit manipulation of intermediate wavelet coefficients of the generator.

## 4.2 QUALITATIVE EVALUATION

We show the qualitative comparisons of inversion and editing results in Figure 4 and Figure 5, respectively. We observed that our model produced more realistic quality of inversion results than baselines, especially when images required more reconstructions of fine-grained details or complex backgrounds. For instance, Hyperstyle (Alaluf et al., 2021b), which refined the weights of the generator per image, failed to reconstruct the out-of-distribution objects, *e.g.*, earrings, and camera. HFGI (Wang et al., 2022b) could generate most of the lost details from the initial inversion via feature fusion, but restored details were close to artifacts, *e.g.*, letters in the background. In contrast, our method was solely capable of reconstructing the details with minimum distortion consistently.

Figure 5 shows the editing results for seven attributes, manipulated by InterFaceGAN directions (Shen et al., 2020) and StyleCLIP (Patashnik et al., 2021). Our model consistently showed the most robust inversion results with high editability, while the baselines failed to edit images and restore details simultaneously. For instance, in the case of InterFaceGAN, the baselines struggled to preserve details, *e.g.*, the hat in the second row or showed low editability, *e.g.*, HFGI in the third

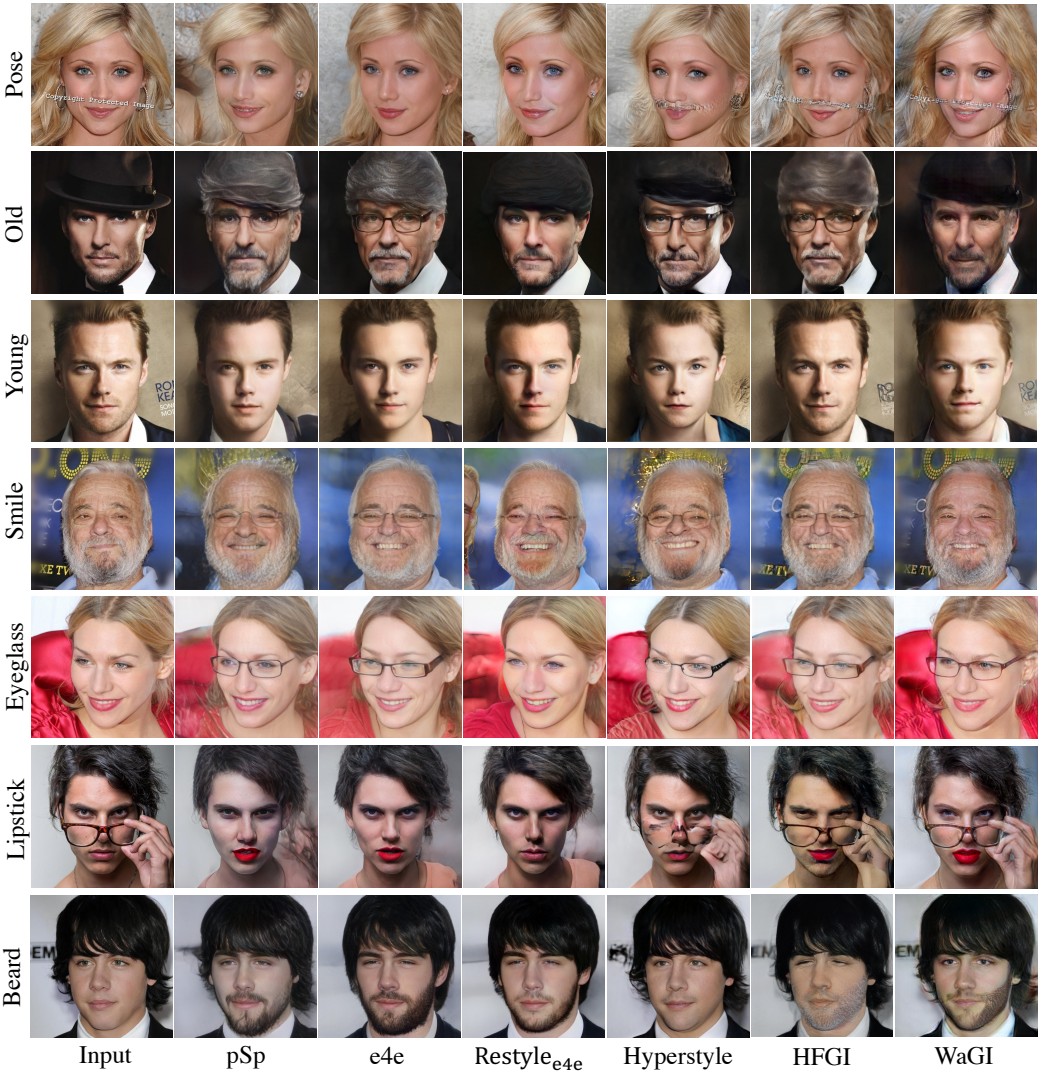

|  | Input | pSp | e4e | Restyle$_{e4e}$ | Hyperstyle | HFGI | WaGI |

Figure 5: **Qualitative comparison between editing results of baselines.** From the first to the fifth rows, we show edited images via InterFaceGAN directions, and editing results for StyleCLIP from the six to seven rows. Both low-rate and high-rate inversion baselines suffer from preserving details, *e.g.*, letters, backgrounds, and hats. HFGI, which relatively restores details among baselines, fails to edit the image in a disentangled way, *e.g.*, distortion of eye shapes for editing with "beard". Our proposed method efficiently restores high-fidelity details with satisfactory editability with highly disentangled editing performance, throughout all various scenarios.

row. In the case of StyleCLIP, similar to InterFaceGAN, the baselines failed to preserve details, *e.g.*, eyeglass at the sixth row, or showed undesirable entanglement, *e.g.*, identity shift of HFGI at the sixth and seventh rows. Overall, both InterFaceGAN and StyleCLIP editing results showed that our model is the most capable of handling the trade-off between reconstruction quality and editability.

## 4.3 ABLATION STUDY

In this section, we analyzed the effectiveness of each component of WaGI, especially for our wavelet loss and wavelet fusion. In Table 2, we quantitatively compared the performance while adding each component we proposed, to the state-of-the-art GAN inversion model, HFGI (Wang et al., 2022a). We compared $L_1$ through $\hat{\Delta}$ for the evaluation of $ADA$, and $L_2$ and SSIM through $\hat{X}$

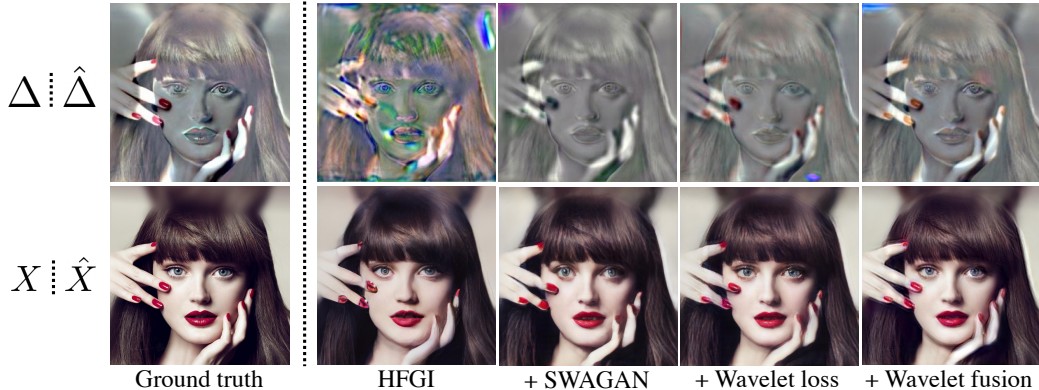

Figure 6: **Qualitative ablation of WaGI.** We compared the performance of $ADA$ and the final inversion from the visualization of $\hat{\Delta}$ and $\hat{X}$, together with $\Delta$ and $X$. While the existing state-of-the-art model introduced severe artifacts for computing $\hat{\Delta}$, our proposed methods reduced artifacts effectively and showed better inversion quality.

| | Configuration | $L_1(\Delta, \hat{\Delta}) \downarrow$ | $L_2(X, \hat{X}) \downarrow$ | $SSIM(X, \hat{X}) \uparrow$ |
|---|---|---|---|---|
| A | HFGI (Wang et al., 2022a) | 0.124 | 0.023 | 0.653 |
| B | + SWAGAN (Gal et al., 2021) | 0.124 | 0.024 | 0.659 |
| C | + Wavelet loss | 0.100 | 0.023 | 0.660 |
| D | + Wavelet fusion | **0.089** | **0.011** | **0.753** |

Table 2: **Quantitative ablation of WaGI.** We compared the performances of $ADA$ and the final inversion by adding each component we proposed. Except for altering generators from StyleGAN to SWAGAN, our proposed methods, *i.e.*, wavelet loss and wavelet fusion, showed remarkable gains in the performance of $ADA$ and the final inversion, respectively.

for the evaluation of the final inversion. Firstly, a simple variation of altering the generator from StyleGAN2 (Karras et al., 2020) to SWAGAN (Gal et al., 2021) (Config B) made only marginal gains for both $ADA$ and the final inversion. We attribute this gain to the characteristics of SWAGAN, which delicately generates the high-frequency more than StyleGAN. In Config C, training with the wavelet loss achieved remarkably lower $L_1(\Delta, \hat{\Delta})$ (about 19%) than training solely with $L_1$. This is compelling that training together with the wavelet loss achieved lower $L_1$ than training solely with $L_1$. From this observation, we can argue that the wavelet loss is not only helpful for preserving the high-frequency sub-band, but also for reducing distortion on the low-frequency sub-band. Moreover, as shown in Figure 6, the wavelet loss is indeed helpful for preserving details, *e.g.*, nail colors, compared to Config B. Finally, when wavelet fusion is combined to C (Config D), *i.e.*, WaGI, there was the apparent gain on both $ADA$ and the final inversion. Until C, though we elaborately calculated high-frequency features, *i.e.*, $\hat{\Delta}$, the model did not effectively transfer it to the generator. However, wavelet fusion enables information transfer to the generator, resulting in the improvement of the inversion. The gains can be shown apparently in the qualitative ways, as in Figure 6.

## 5 CONCLUSION

Recent high-rate GAN inversion methods focus on preserving image-wise details but still suffer from the low-frequency bias. We point out that the existing methods are biased on low-frequency sub-band, in both structural and training aspects. To overcome this, we proposed a novel GAN inversion, named WaGI, which explicitly handles the wavelet coefficients of the high-frequency sub-band via wavelet loss and wavelet fusion. We demonstrated that WaGI achieved the best performances among the state-of-the-art GAN inversion methods. Moreover, we explored the effectiveness of each of the proposed methods through the elaborate ablation study. Since our framework is simple and can be simply reproduced with our released code, we look forward to its wide usage in future work.

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

# A APPENDIX

## A.1 PROOFS FOR THE THEOREM

In this section, we show the proof for the proposed theorem and lemma in the main paper;

**Theorem 1.** *The following equation holds when $\lambda_f = 1$, $\forall f \in \mathbb{F}_l \cup \mathbb{F}_h$:*

$$L_2(I_1, I_2) = \sum_{f \in \mathbb{F}_l \cup \mathbb{F}_h} \lambda_f L_{2,f}(I_1, I_2). \tag{10}$$

*Proof.* Let $I_1 = (a_{ij}) \in \mathbb{R}^{m \times n}$, and $I_2 = (b_{ij}) \in \mathbb{R}^{m \times n}$. And for the simplicity of the notation, we define $c_{i,j} = a_{i,j} - b_{i,j}$.

Then,

$$L_2(I_1, I_2) = \frac{1}{mn} \sum_{i \in \{0,1,...m\}} \sum_{j \in \{0,1,...n\}} (c_{ij})^2.$$

Let $m' = [\frac{m}{2}]$, $n' = [\frac{n}{2}]$. Then, we can denote $L_{2, f \in \mathbb{F}_l \cup \mathbb{F}_h}$ as below;

$$L_{2,LL}(I_1, I_2) = \frac{1}{m'n'} \sum_{i \in \{0,1,...m'\}} \sum_{j \in \{0,1,...n'\}} (c_{2i+1,2j+1} + c_{2i+1,2j} + c_{2i,2j+1} + c_{2i,2j})^2,$$

$$L_{2,LH}(I_1, I_2) = \frac{1}{m'n'} \sum_{i \in \{0,1,...m'\}} \sum_{j \in \{0,1,...n'\}} (-c_{2i+1,2j+1} - c_{2i+1,2j} + c_{2i,2j+1} + c_{2i,2j})^2,$$

$$L_{2,HL}(I_1, I_2) = \frac{1}{m'n'} \sum_{i \in \{0,1,...m'\}} \sum_{j \in \{0,1,...n'\}} (-c_{2i+1,2j+1} + c_{2i+1,2j} - c_{2i,2j+1} + c_{2i,2j})^2,$$

$$L_{2,HH}(I_1, I_2) = \frac{1}{m'n'} \sum_{i \in \{0,1,...m'\}} \sum_{j \in \{0,1,...n'\}} (c_{2i+1,2j+1} - c_{2i+1,2j} - c_{2i,2j+1} + c_{2i,2j})^2.$$

We can rewrite $L_2(I_1, I_2)$ as;

$$L_2(I_1, I_2) = \frac{4}{m'n'} \sum_{i \in \{0,1,...m'\}} \sum_{j \in \{0,1,...n'\}} c_{2i+1,2j+1}^2 + c_{2i+1,2j}^2 + c_{2i,2j+1}^2 + c_{2i,2j}^2.$$

We use the following identical equation, which holds for $\forall x, y, z, w \in \mathbb{R}$;

$$(x+y+z+w)^2 + (-x-y+z+w)^2 + (-x+y-z+w)^2 + (x-y-z+w)^2 = 4(x^2+y^2+z^2+w^2).$$

We can obtain the followings;

$$\sum_{f \in \mathbb{F}_l \cup \mathbb{F}_h} L_{2,f}(I_1, I_2) = \frac{1}{m'n'} \sum_{i \in \{0,1,...m'\}} \sum_{j \in \{0,1,...n'\}} 4(c_{2i+1,2j+1}^2 + c_{2i+1,2j}^2 + c_{2i,2j+1}^2 + c_{2i,2j}^2).$$

$$\therefore L_2(I_1, I_2) = \sum_{f \in \mathbb{F}_l \cup \mathbb{F}_h} 1 \cdot L_{2,f}(I_1, I_2).$$

$\square$

**Lemma 1.1.** *When the distributions of pixel-wise differences between $I_1$ and $I_2$ are i.i.d., and follow $\mathcal{N}(\mu, \sigma^2)$ with $\mu \approx 0$, the following equation holds when $\lambda_f = \frac{1}{4}$, $\forall f \in \mathbb{F}_l \cup \mathbb{F}_h$:*

$$\log \mathbb{E}[L_1(I_1, I_2)] + C = \sum_{f \in \mathbb{F}_l \cup \mathbb{F}_h} \lambda_f \log \mathbb{E}[L_{1,f}(I_1, I_2)], \tag{11}$$

*where $C$ is a constant.*

*Proof.* Similar with proving Theorem 1, we can derive followings;

$$L_1(I_1, I_2) = \frac{4}{m'n'} \sum_{i \in \{0,1,\dots m'\}} \sum_{j \in \{0,1,\dots n'\}} |c_{2i+1,2j+1}| + |c_{2i+1,2j}| + |c_{2i,2j+1}| + |c_{2i,2j}|,$$

$$L_{1,LL}(I_1, I_2) = \frac{1}{m'n'} \sum_{i \in \{0,1,\dots m'\}} \sum_{j \in \{0,1,\dots n'\}} |c_{2i+1,2j+1} + c_{2i+1,2j} + c_{2i,2j+1} + c_{2i,2j}|,$$

$$L_{1,LH}(I_1, I_2) = \frac{1}{m'n'} \sum_{i \in \{0,1,\dots m'\}} \sum_{j \in \{0,1,\dots n'\}} |-c_{2i+1,2j+1} - c_{2i+1,2j} + c_{2i,2j+1} + c_{2i,2j}|,$$

$$L_{1,HL}(I_1, I_2) = \frac{1}{m'n'} \sum_{i \in \{0,1,\dots m'\}} \sum_{j \in \{0,1,\dots n'\}} |-c_{2i+1,2j+1} + c_{2i+1,2j} - c_{2i,2j+1} + c_{2i,2j}|,$$

$$L_{1,HH}(I_1, I_2) = \frac{1}{m'n'} \sum_{i \in \{0,1,\dots m'\}} \sum_{j \in \{0,1,\dots n'\}} |c_{2i+1,2j+1} - c_{2i+1,2j} - c_{2i,2j+1} + c_{2i,2j}|.$$

Using $c_{i,j} \sim \mathcal{N}(\mu, \sigma^2)$, we can obtain followings:

$$(c_{2i+1,2j+1} + c_{2i+1,2j} + c_{2i,2j+1} + c_{2i,2j}) \sim \mathcal{N}(4\mu, 4\sigma^2),$$

$$(-c_{2i+1,2j+1} - c_{2i+1,2j} + c_{2i,2j+1} + c_{2i,2j}) \sim \mathcal{N}(0, 4\sigma^2),$$

$$(-c_{2i+1,2j+1} + c_{2i+1,2j} - c_{2i,2j+1} + c_{2i,2j}) \sim \mathcal{N}(0, 4\sigma^2),$$

$$(c_{2i+1,2j+1} - c_{2i+1,2j} - c_{2i,2j+1} + c_{2i,2j}) \sim \mathcal{N}(0, 4\sigma^2).$$

According to the properties of half-normal distribution, for $p \sim \mathcal{N}(\mu, \sigma^2)$,

$$\mathbb{E}[|p|] = \sigma\sqrt{\frac{2}{\pi}} e^{-\frac{\mu^2}{2\sigma^2}} + \mu \cdot \text{erf}(\frac{\mu}{\sqrt{(2\sigma^2)}}), \text{ where } \text{erf}(x) = \int_0^x e^{-t^2} dt.$$

Consequently,

$$\mathbb{E}[|c_{2i+1,2j+1}| + |c_{2i+1,2j}| + |c_{2i,2j+1}| + |c_{2i,2j}|] = 4\sigma\sqrt{\frac{2}{\pi}} e^{-\frac{\mu^2}{2\sigma^2}} + 4\mu \cdot \text{erf}(\frac{\mu}{\sqrt{(2\sigma^2)}}),$$

$$\mathbb{E}[|c_{2i+1,2j+1} + c_{2i+1,2j} + c_{2i,2j+1} + c_{2i,2j}|] = 2\sigma\sqrt{\frac{2}{\pi}} e^{-4 \cdot \frac{\mu^2}{2\sigma^2}} + 4\mu \cdot \text{erf}(\frac{2\mu}{\sqrt{(2\sigma^2)}}),$$

$$\mathbb{E}[|-c_{2i+1,2j+1} - c_{2i+1,2j} + c_{2i,2j+1} + c_{2i,2j}|] = 2\sigma\sqrt{\frac{2}{\pi}},$$

$$\mathbb{E}[|-c_{2i+1,2j+1} + c_{2i+1,2j} - c_{2i,2j+1} + c_{2i,2j}|] = 2\sigma\sqrt{\frac{2}{\pi}},$$

$$\mathbb{E}[|c_{2i+1,2j+1} - c_{2i+1,2j} - c_{2i,2j+1} + c_{2i,2j}|] = 2\sigma\sqrt{\frac{2}{\pi}}.$$

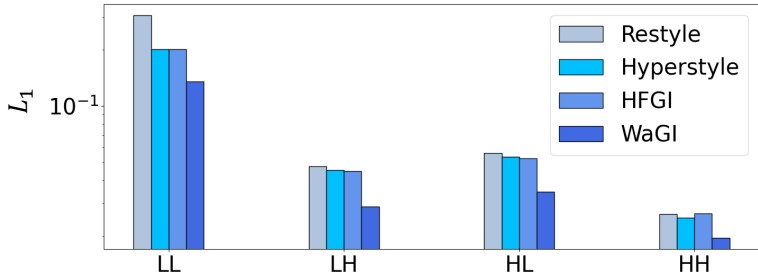

Figure 7: **Comparison of $L_1$ of the wavelet coefficients.** We plot the average $L_1$ of each wavelet coefficient between CelebA-HQ test images and corresponding inverted images by various state-of-the-art inversion models. Due to the significant gap between $L_{1,LL}$ and the rest (about 30 times), we display the losses with the logarithmic scale for better visualization.

Using the condition that $c_{i,j}$s are *i.i.d.*,

$$\mathbb{E}[L_1(I_1, I_2)] = \frac{4}{m'n'} \cdot m'n' \cdot (4\sigma\sqrt{\frac{2}{\pi}}e^{-\frac{\mu^2}{2\sigma^2}} + 4\mu \cdot \text{erf}(\frac{2\mu}{\sqrt{2\sigma^2}}))$$

$$= 16\sigma\sqrt{\frac{2}{\pi}}e^{-\frac{\mu^2}{2\sigma^2}} + 16\mu \cdot \text{erf}(\frac{2\mu}{\sqrt{2\sigma^2}}),$$

$$\mathbb{E}[L_{1,LL}(I_1, I_2)] = \frac{1}{m'n'} \cdot m'n' \cdot (2\sigma\sqrt{\frac{2}{\pi}}e^{-4\cdot\frac{\mu^2}{2\sigma^2}} + 4\mu \cdot \text{erf}(\frac{2\mu}{\sqrt{(2\sigma^2)}}))$$

$$= 2\sigma\sqrt{\frac{2}{\pi}}e^{-4\cdot\frac{\mu^2}{2\sigma^2}} + 4\mu \cdot \text{erf}(\frac{\mu}{\sqrt{(2\sigma^2)}}),$$

$$\mathbb{E}[L_{1,LH}(I_1, I_2)] = \mathbb{E}[L_{1,HL}(I_1, I_2)] = \mathbb{E}[L_{1,HH}(I_1, I_2)] = \frac{1}{m'n'} \cdot m'n' \cdot 2\sigma\sqrt{\frac{2}{\pi}} = 2\sigma\sqrt{\frac{2}{\pi}}.$$

Since $\mu \approx 0$, $\mu \cdot \text{erf}(\frac{\mu}{\sqrt{2\sigma^2}}) \approx 0$. Consequently,

$$\log \mathbb{E}[L_1(I_1, I_2)] = \log 16 + \log(\sigma\sqrt{\frac{2}{\pi}}) - \frac{\mu^2}{2\sigma^2},$$

$$\log \mathbb{E}[L_{1,LL}(I_1, I_2)] = \log 2 + \log(\sigma\sqrt{\frac{2}{\pi}}) - 4 \cdot \frac{\mu^2}{2\sigma^2},$$

$$\log \mathbb{E}[L_{1,LH}(I_1, I_2)] = \log \mathbb{E}[L_{1,HL}(I_1, I_2)] = \log \mathbb{E}[L_{1,HH}(I_1, I_2)] = \log 2 + \log(\sigma\sqrt{\frac{2}{\pi}}).$$

$$\therefore \log \mathbb{E}[L_1(I_1, I_2)] = \sum_{f \in \mathbb{F}_l \cup \mathbb{F}_h} \frac{1}{4} \log \mathbb{E}[L_{1,f}(I_1, I_2)] + C.$$

$\square$

Similar with $L_2$, $L_1$ seems a fair loss without the frequency bias, which reflects $L_{1,f \in \mathbb{F}_l \cup \mathbb{F}_h}$ with same weights. However, as shown in Figure 7, we empirically find that $L_{1,LL}$ is around 30 times larger than $L_{1,f \in \mathbb{F}_h}$ in case of HyperStyle and HFGI. This leads to the biased training, which results in apparent decrease of $L_{1,LL}$, but almost no gain, or even increment of $L_{1,f \in \mathbb{F}_h}$, compared to Restyle. Consequently, we argue that $L_1$ contains the low-frequency bias, and needs the wavelet loss to avoid it.

## A.2 FREQUENCY BIAS

### A.2.1 GENERATOR TRAINING WITH SPECTRAL LOSS

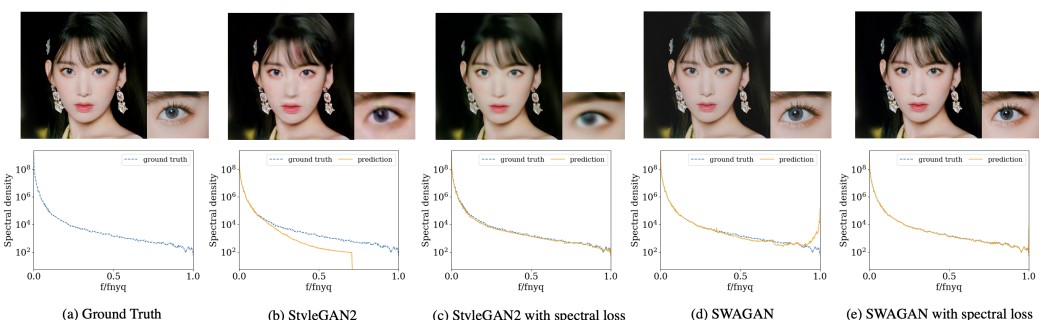

Figure 8: **Regression of a single image (top row) and spectral density plot of ground truth image and generated images (bottom row) trained with/without additional spectral loss.** Here, we used the spectral loss introduced in Schwarz et al. (2021). For both StyleGAN2 and SWAGAN generators, the additional spectral loss induced artifacts to coercively match the frequency distribution. We recommend you zoom in to carefully observe the reconstructed details.

Previous works (Schwarz et al., 2021; Jiang et al., 2021; Durall et al., 2020) each propose an objective function to precisely learn the frequency distribution of the training data, which we comprehensively named them as *spectral loss*. Jiang et al. (2021) designed a spectral loss function that measures the distance between fake and real images in the frequency domain that captures both amplitude and phase information. Durall et al. (2020) proposed a spectral loss that measures the binary cross entropy between the azimuthal integration over power spectrum of fake and real images. Schwarz et al. (2021) used a simple $L_2$ loss between the logarithm of the azimuthal average over power spectrum in normalized polar coordinates, *i.e.*reduced spectrum, of fake and real images. We adopted the spectral loss term of Schwarz et al. (2021) for our experiment :

$$L_S = \frac{1}{H/\sqrt{2}} \sum_{k=0}^{H/\sqrt{2}-1} \| \log\left(\tilde{S}(G(z))\right)[k] - \log\left(\tilde{S}(\mathbf{I})\right)[k]\|_2^2,$$ (12)

where $\tilde{S}$ is the reduced spectrum, $G(z)$ is the generated image, and $\mathbf{I}$ is the ground truth real image.

Here, we conducted a single-image reconstruction task, which is widely done (Gal et al., 2021; Schwarz et al., 2021) to investigate the effectiveness of explicit frequency matching in refining high-fidelity details. For StyleGAN2 (Karras et al., 2020) and SWAGAN (Gal et al., 2021) generator, we used the latent optimization (Karras et al., 2019) method to reconstruct a single image, each with and without the spectral loss. All images are generated to resolution $512 \times 512$, with the weight of spectrum loss $\times 0.1$ of the original $L_2$ loss.

Figure 8 shows the reconstructed images and spectral density plots for each case. As seen in Figure 8(a), the spectrum of a natural image follows a exponentially decay. Using $L_2$ singularly made both StyleGAN2 and SWAGAN generators overfit to the mostly existing low-frequency distribution. (b) StyleGAN2 struggled to learn the high-fidelity details, creating an unrealistic image. (d) SWAGAN was capable of fitting most of the high-frequency parts, except created some excessive high frequency noise due to checkerboard patterns. Though utilizing the spectral loss for both generators (c,e) exquisitely matched all frequency distribution, qualitative results were degraded. Matching the frequency induced unwanted artifacts to the images, and caused the degradation. Due to the absence of the spatial information, the loss based on the spectral density inherently cannot robustly reconstruct high-frequency details. Comparably, our wavelet loss minimizes the $L_1$ distance of high-frequency bands in the spatial frequency domain, restoring meaningful high-fidelity features.

### A.2.2 INFORMATION IN SUB-BAND OF IMAGES

In Section 3.2, we designed the multi-level wavelet loss to cover broader frequency ranges than $f_{nyq}/2 \sim f_{nyq}$. In Figure 9, we show the results of the inverse wavelet transform by omitting the

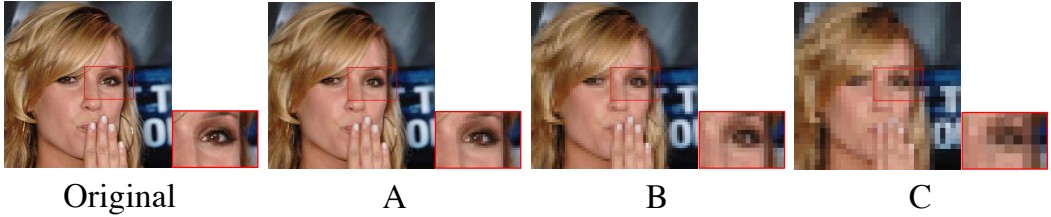

| Original | A | B | C |

Figure 9: **Inverse wavelet transform results by omitting various wavelet sub-bands.** To check the qualitative image details in each sub-band, we remove the wavelet coefficients between $f_{nyq}/2$ $\sim f_{nyq}$ (Config A), $f_{nyq}/2^2 \sim f_{nyq}/2$ (Config B), and $f_{nyq}/2^3 \sim f_{nyq}/2^2$ (Config C). From A to B, severe degradation of visible image details do not occur. However, for B to C or C to D, majority of image details are degraded.

wavelet coefficients between $f_{nyq}/2 \sim f_{nyq}$ (Config A), $f_{nyq}/2^2 \sim f_{nyq}/2$ (Config B), and $f_{nyq}/2^3$ $\sim f_{nyq}/2^2$ (Config C). Though A removes the highest frequency sub-bands, *i.e.*, $f_{nyq}/2 \sim f_{nyq}$, among all configs, we cannot find visible degradation of image details. In other words, information in the sub-band $f_{nyq}/2 \sim f_{nyq}$ is mostly higher than the visible image details. Since the firstly proposed wavelet loss in Equation 5 only covers the sub-band $f_{nyq}/2 \sim f_{nyq}$, we should extend the range of sub-bands to effectively preserve the visible details. Consequently, we propose a $K$-level wavelet loss, which enables covering the sub-band $f_{nyq}/2^{K+1} \sim f_{nyq}$.

## A.3 Experimental Details

### A.3.1 Training Details

In our experiments, we implement our experiments based on the pytorch-version code [2] for SWA-GAN (Gal et al., 2021). We converted the weights of pretrained SWAGAN generator checkpoint from the official tensorflow code[3] to a pytorch version. We trained our model on a single GPU and took only 6 hours for the validation loss to saturate, whereas other StyleGAN2-based baselines required more than 2 days of training time.

Here, we explain the details of our reconstruction loss terms: $L_2$, $L_{id}$, and $L_{LPIPS}$. We leverage $L_2$, as it is most effective in keeping the generated image similar to the original image pixel-wise. $L_{id}$ is an identity loss function defined as :

$$L_{id} = 1- < R(G_0(w)), R(\mathbf{I}) >, \qquad (13)$$

where $R$ is the pre-trained ArcFace (Deng et al., 2021) model, and $\mathbf{I}$ is the ground truth image. $L_{id}$ minimizes the cosine distance between two face images to preserve the identity . LPIPS (Zhang et al., 2018) enhances the perceptual quality of the image via minimizing the feature space of generated images and the feature space of ImageNet (Deng et al., 2009) pre-trained network. For training, we used weights $\lambda_{L_2}=1$, $\lambda_{id}=0.1$, $\lambda_{LPIPS}=0.8$, respectively.

### A.3.2 Dataset Description

**Flickr-Faces-HQ (FFHQ) dataset** (Karras et al., 2019). Our model and all baselines are trained with FFHQ, a well-aligned human face dataset with 70,000 images of resolution $1024 \times 1024$. FFHQ dataset is widely used for training various unconditional generators (Karras et al., 2019; 2020; 2021), and GAN inversion models (Richardson et al., 2021; Tov et al., 2021; Alaluf et al., 2021a;b; Moon & Park, 2022; Wang et al., 2022a). All of the baselines we used in the paper use the FFHQ dataset for training, which enables a fair comparison.

**CelebA-HQ (FFHQ) dataset** CelebA-HQ dataset contains 30,000 human facial images of resolution $1024 \times 1024$, together with the segmentation masks. Among 30,000 images, around 2,800 images are denoted as the test dataset. We use the official split for the test dataset, and evaluate every baseline and our model with all images in the test dataset.

---

[2]https://github.com/rosinality/stylegan2-pytorch
[3]https://github.com/rinongal/swagan

### A.3.3 Baseline Models Description

In this section, we describe the existing GAN inversion baselines, which we used for comparison in Section 4. We exclude the model which needs image-wise optimization, such as Image2StyleGAN Abdal et al. (2019) or Pivotal Tuning Roich et al. (2021).

**pSp** pixel2Style2pixel ($pSp$) adopts pyramid (Lin et al., 2017) network for the encoder-based GAN inversion. $pSp$ achieves the state-of-the-art performance among encoder-based inversion models at the time. Moreover, $pSp$ shows the various adaptation of the encoder model to the various tasks using StyleGAN, such as image inpainting, face frontalization, or super-resolution.

**e4e** encoder4editing ($e4e$) proposes the existence of the trade-off between distortion and the perception-editability of the image inversion. In the other words, $e4e$ proposes that the existing GAN inversion models which focus on lowering distortion, sacrifice the perceptual quality of inverted images, and the robustness on the editing scenario. $e4e$ suggests that maintaining the latent close to the original StyleGAN latent space, *i.e.*, $W$, enables the inverted image to have high perceptual quality and editability. To this end, $e4e$ proposes additional training loss terms to keep the latent close to $W$ space. Though distortion of $pSp$ is lower than $e4e$, $e4e$ shows apparently higher perceptual quality and editabilty than $pSp$.

**Restyle** Restyle suggests that a single feed-forward operation of existing encoder-based GAN inversion models, *i.e.*, $pSp$ and $e4e$, is not enough to utilize every detail in the image. To overcome this, Restyle proposes an iterative refinement scheme, which infers the latent with feed-forward-based iterative calculation. The lowest distortion that Restyle achieves among encoder-based GAN inversion models shows the effectiveness of the iterative refinement scheme. Moreover, the iterative refinement scheme can be adapted to both $pSp$ and $e4e$, which enables constructing models that have strengths in lowering distortion, or high perceptual quality-editability, respectively. To the best of our knowledge, Restyle$_{pSp}$ achieves the lowest distortion among encoder-based models which do not use generator-tuning method[4]. Since we utilize baselines that achieves lower distortion than Restyle$_{pSp}$, *i.e.*, HyperStyle and HFGI, we only use Restyle$_{e4e}$ to evaluate its high editability.

**HyperStyle** To make a further improvement from Restyle, Pivotal Tuning (Roich et al., 2021) uses the input-wise generator tuning. However, this is extremely time-consuming, and inconvenient in that it requires separate generators per every input image. To overcome this, HyperStyle adopts HyperNetwork (Ha et al., 2016), which enables tuning the convolutional weights of pre-trained StyleGAN only with the feed-forward calculation. Starting from the latent obtained by $e4e$, HyperStyle iteratively refines the generator to reconstruct the original image with the fixed latent. HyperStyle achieves the lowest distortion among encoder-based GAN inversion models at the time.

**HFGI** HFGI points out the limitation of the low-rate inversion methods and argues that encoders should adopt larger dimensions of tensors to transfer high-fidelity image-wise details. To achieve this, HFGI adapts feature fusion, which enables mixing the original StyleGAN feature with the feature obtained by the image-wise details. To the best of our knowledge, HFGI achieves the lowest distortion among every GAN inversion method, even including Pivotal Tuning, except our model.

### A.3.4 Ablation studies

**Choice of fusion layer.** We additionally provide both quantitative and qualitative ablation results for the inversion performance of WaGI with fusion in different layers. Note that in our main experiment, we apply feature fusion in layers $\ell_f = 7$ and $9$, and wavelet fusion in layer $\ell_w = 11$. Each layer corresponds to fusion of spatial features with resolution $64 \times 64$ and $128 \times 128$, and wavelet coefficients of dimension $w \in \mathbb{R}^{12 \times 128 \times 128}$. From the quantitative results in Table 3, we observed that the feature fusion on two layers $\ell_f = 7$ and $9$ showed better reconstruction accuracy than on a single layer $\ell_f = 7$. Additionally, wavelet fusion in lower layers ($\ell_w < 11$) was not sufficient enough to preserve the high-fidelity details, especially in the high-frequency region *i.e.* $L_{wave}$. Wavelet fusion in the higher layer ($\ell_w = 13$) also degraded the inversion performance, which can be more carefully observed in Figure 10.

---

[4]IntereStyle (Moon & Park, 2022) achieves lower distortion on the *interest region* than Restyle$_{pSp}$, but not for the whole image region.

| Feature fusion | Wavelet fusion | $L_2 \downarrow$ | $L_{wave} \downarrow$ | LPIPS $\downarrow$ | SSIM $\uparrow$ | ID sim $\uparrow$ |
|---|---|---|---|---|---|---|
| $\ell_f = 7$ | $\ell_w = 7$ | 0.028 | 0.359 | 0.365 | 0.667 | 0.796 |
| | $\ell_w = 9$ | 0.026 | 0.356 | 0.362 | 0.701 | 0.830 |
| | $\ell_w = 11$ | 0.026 | 0.325 | 0.364 | 0.727 | 0.847 |
| | $\ell_w = 13$ | 0.024 | 0.314 | 0.366 | 0.727 | 0.845 |
| $\ell_f = 7$ and 9 | $\ell_w = 7$ | 0.020 | 0.327 | 0.346 | 0.711 | 0.849 |
| | $\ell_w = 9$ | 0.016 | 0.289 | 0.330 | 0.724 | 0.880 |
| | $\ell_w = 11$ | **0.011** | **0.230** | **0.277** | **0.753** | **0.906** |
| | $\ell_w = 13$ | 0.020 | 0.307 | 0.342 | 0.722 | 0.861 |

Table 3: **Ablation of the fusion layers for WaGI.** We compared the inversion performance of WaGI with feature and wavelet fusion in different layers. Feature fusion on layers $\ell_f = 7$ and 9, and wavelet fusion on layer $\ell_w = 11$ consistently showed the highest fidelity and reconstruction quality among all scenarios.

Figure 10 shows the inverted images for each scenario in Table 3. It is noticeable that fusion in a single layer (a)-(d) failed to retain high-frequency details like the hand and hair texture. Comparably, in the case of multi-layer feature fusion (e)-(h), inverted images reconstructed more high-frequency details. Yet, wavelet fusion in the lower layers (e), (g), and higher layer (h) generated unwanted distortions, which eventually degraded the image fidelity. Overall, our scenario (g) empirically showed the most promising reconstruction quality, generating realistic images with the least distortion.

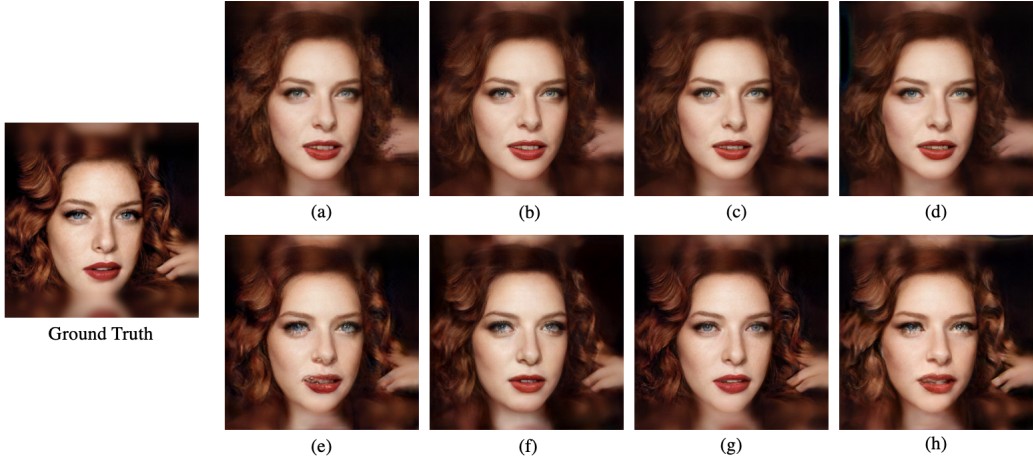

Figure 10: **Quantitative comparison of WaGI inversion with fusion in different layers.** Each image represents the inversion results for each scenario in Table 3. The first row (a)-(d) displays inverted images with feature fusion in a single layer $\ell_f = 7$, with wavelet fusion in layer $\ell_w = 7$, $\ell_w = 9$, $\ell_w = 11$, and $\ell_w = 13$, respectively. The second row (e)-(h) displays inverted images with feature fusion in multi-layers $\ell_f = 7$ and 9, with wavelet fusion in layer $\ell_w = 7$, $\ell_w = 9$, $\ell_w = 11$, and $\ell_w = 13$, respectively. We recommend you zoom in for a careful look into the details.

**Design of fusion methods.** To prove the effectiveness of the wavelet fusion, we compared the performance of WaGI with the model which uses the feature fusion, proposed in HFGI (Wang et al., 2022a), instead of the wavelet fusion in the same resolution layer. In Table 4, we compared the performance of models with the following four settings: The original HFGI which uses feature fusion at $l_f = 7$, HFGI with additional feature fusion at $l_f = 9$ and 11, WaGI with the feature fusion at $l_f = 7$, 9, and 11, and the original WaGI which uses the feature fusion at $l_f = 7$ and 9, and the wavelet fusion at $l_w = 11$. First, simply adding the feature fusion to the higher layer is not helpful for improving the model. If we change it to the WaGI method, *i.e.*, change the generator and add the wavelet loss, the performance significantly improved. And after changing the feature fusion at the

| Model | $L_2 \downarrow$ | $L_{wave} \downarrow$ | SSIM $\uparrow$ | ID sim $\uparrow$ |
|---|---|---|---|---|
| HFGI ($\ell_f = 7$) | 0.023 | 0.351 | 0.661 | 0.864 |
| HFGI ($\ell_f = 7, 9, 11$) | 0.036 | 0.377 | 0.704 | 0.795 |
| WaGI ($\ell_f = 7, 9, 11$) | 0.017 | 0.302 | 0.699 | 0.873 |
| **WaGI** ($\ell_f = 7, 9$ **and** $\ell_w = 11$) | **0.011** | **0.230** | **0.753** | **0.906** |

Table 4: **Ablation of the fusion methods for WaGI.** We compared the inversion performance of WaGI with the model which uses wavelet fusion instead of feature fusion. Though changing all the fusion methods with the feature fusion achieves better results than HFGI, still it shows a big performance degradation compared to WaGI.

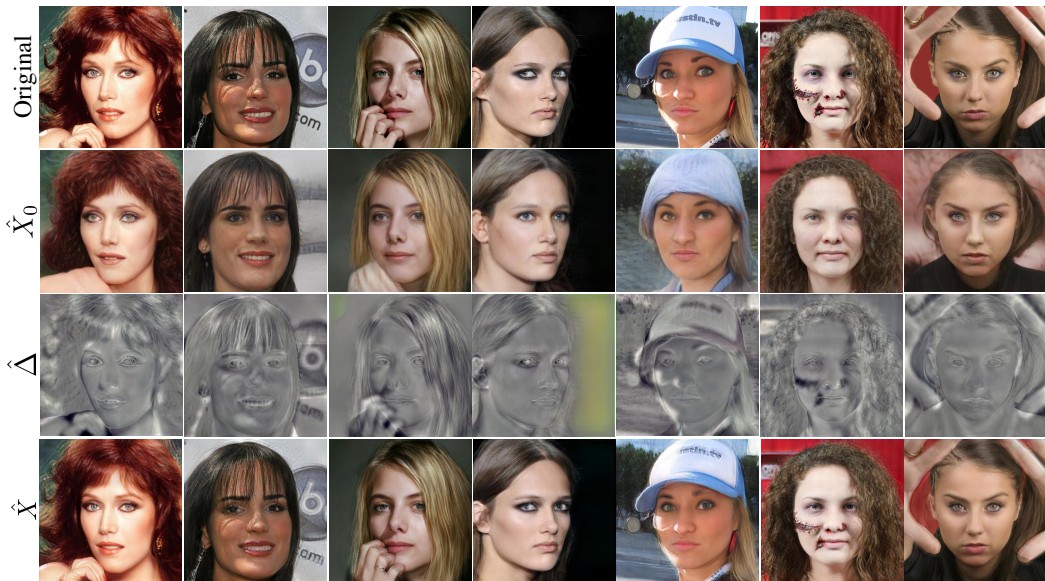

Figure 11: **Visualization of intermediate features during inversion.** We visualize the intermediate output of our model for the inversion scenario. $\hat{\Delta}$ indeed preserves image-specific details.

$11 - th$ layer, *i.e.*, WaGI, the performance remarkably improved and achieves state-of-the-art results on various metrics.

A.3.5 ADDITIONAL EXPERIMENTAL RESULTS.

In this section, we show the additional experimental results of WaGI. First, in Figure 11, we showed the visualization results of intermediate features in the inversion scenario. We indeed find that $\hat{\Delta}$ reflects the image-wise details precisely, which results in the robust reconstruction in $\hat{X}$. Despite the images in Figure 11 containing extreme out-of-distribution components, *e.g.*, complex backgrounds, fingers, and big scars on the face, our model preserves every detail consistently. In Figure 12, we provide an extensive comparison between baselines and our model on both inversion and editing scenarios. While baselines omitted or deformed the image-wise details, our model robustly preserve details while maintaining high editability.

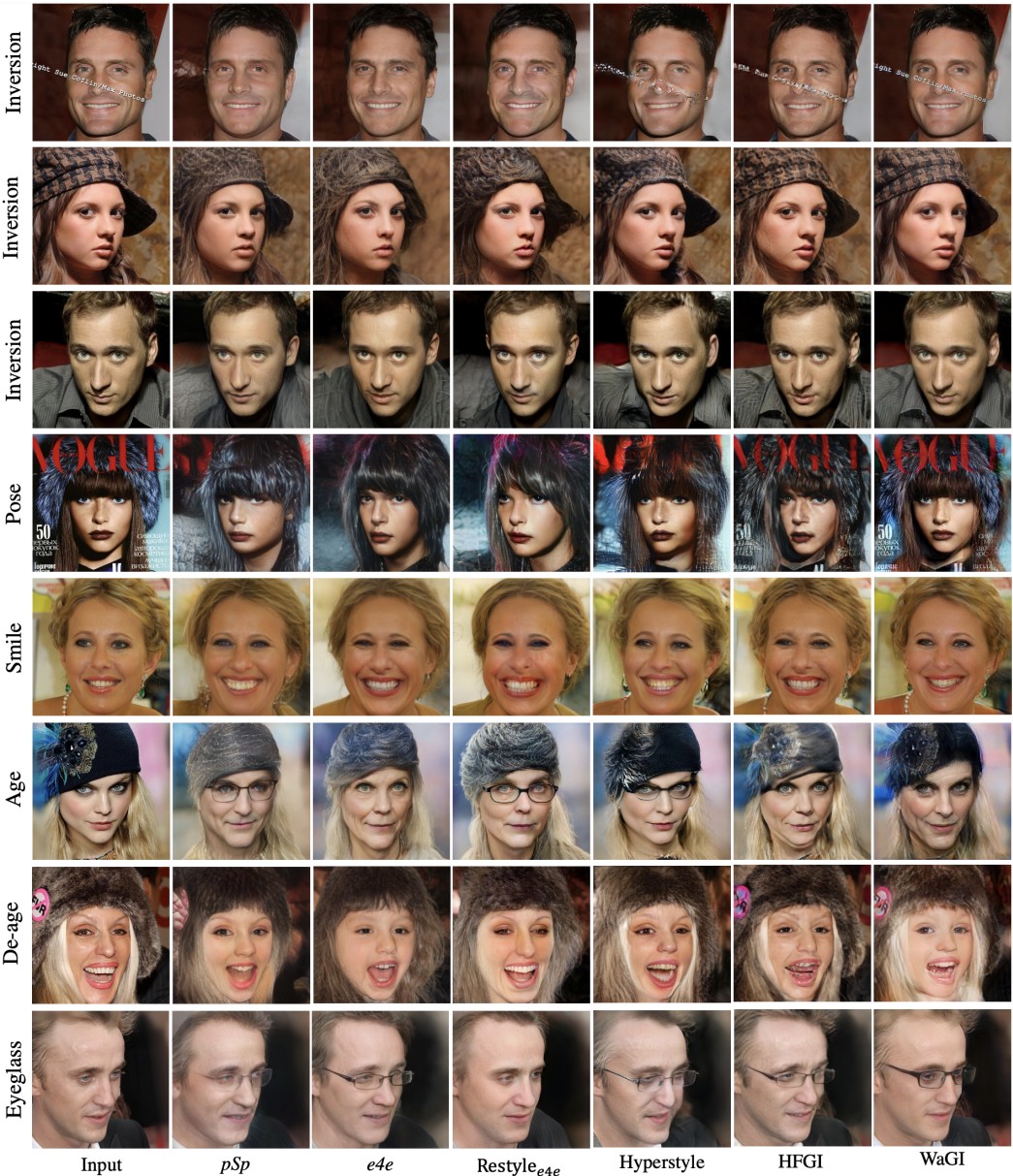

Figure 12: **Qualitative comparison between inversion and editing results of baselines.** From the first to the third rows, we show the inverted images. The inverted image from WaGI preserves the most high-frequency details like the texture of clothes and skin. From the fourth and eighth rows, we show edited images of various attributes. WaGI is the only method capable of editing images without loss of high-frequency information.

