# OpenReview forum: "WaGI: Wavelet-based GAN Inversion for Preserving High-Frequency Image Details"
_ICLR.cc/2023/Conference — Submitted to ICLR 2023_

### Official Review · Reviewer_Xb3x · 2022-10-19

**Confidence:** 3
**Correctness:** 3
**Technical Novelty And Significance:** 3
**Empirical Novelty And Significance:** 3
**Recommendation:** 8

**Clarity, Quality, Novelty And Reproducibility:**

The paper is written in a clear and easy to follow fashion. The authors claim that they will provide code and models for the public community. The paper is an incremental work combining ideas from two previous papers, namely, SWAGAN and HFGI, yet the implementation is not trivial and the results look good.

**Details Of Ethics Concerns:**

no ethics concerns

**Strength And Weaknesses:**

## Strength
The paper is a natural extension of the HFGI and SWAGAN papers. The authors addressed critical points, in the continuations of the ideas presented in the mentioned papers to extended the HFGI method.
In qualitative evaluation the method show better results on the inversion task, especially on text, which is usually a high frequency signal. In addition, it seems like this method is better than HFGI for editing tasks (visually).

## Weaknesses
The authors suggest wavelet losses for high-frequency details reconstructions. Another option to try is to use the HFGI method while adding additional latent layers. The HFGI method fuses only the earlier layer of G_0 and therefore is unlikely that high-frequency features will be preserved.

The authors did not show ablation study on which of the G_0 layers they use for fusion.

The authors write in the abstract that they proved the loss used for others models is biased to the low frequency features. First, the analysis show the loss is unbiased. Second, I'm not sure if the empirical evidence they provide in order to show the loss is indeed biased in practice, is sufficiently convincing.

**Summary Of The Paper:**

The paper presents and extension of the HFGI method for GAN-Inversion, named WaGI. The author propose to replace the StyleGAN generator with a SWAGAN generator, and apply Wavelet loss on the HFGI ADA's module. In addition, they propose a Wavelet fusion method to fuse the into the latent \w. The authors suggest that previous methods fail to preserve high-frequency details where their method   succeed. The authors prove that the widely used loss term in GAN inversions is biased.

**Summary Of The Review:**

I support the acceptance of the paper due to the importance to the community, the nice implementation details and nice results.

---

> ### Author Response · Authors · 2022-11-15
> **Response to Reviewer Xb3x**
>
> We are sincerely thankful to every reviewer for reading our research narrowly and giving us thoughtful feedback.
> We are glad to give the impression to all reviewers that our method is reasonable and effective.
> We carefully respond to each of the concerns and questions, together with a revised manuscript that reflects all comments.
> ___
> **Q1. Ablation for using HFGI method while adding additional latent layers.**
>
> Thanks for the meaningful comment.
> We added the ablation studies related to using the HFGI method instead of the wavelet fusion in Appendix A.3.4 of the revised manuscript.
> Here, we briefly show the results of the additional ablation studies you are curious about.
> First, we tried adding the feature fusion layers on the original HFGI method to check the effectiveness of the feature fusion on higher resolution layers.
> Then, we show the WaGI result of using the feature fusion method proposed in HFGI instead of the wavelet fusion at the $11^{th}$ layer, i.e., the 256-resolution layer in SWAGAN.
> Among all the variants, WaGI shows the best performance on various metrics, with remarkable gap with the others.
> Model | $L_2 \downarrow$ | $L_{wave}\downarrow$ | SSIM $\uparrow$ |ID sim$\uparrow$
> -------|-------|--------|-------|--------
> HFGI ($\ell_f=7$)  | 0.023 | 0.351 | 0.661 | 0.864
> HFGI ($\ell_f=7, 9, 11$)  | 0.036 | 0.377 | 0.704 | 0.795
> WaGI ($\ell_f=7, 9, 11$)   | 0.017 | 0.302 | 0.699 | 0.873
> **WaGI** ($\ell_f=7, 9$  **and**   $\ell_w=11$)   |  **0.011** | **0.230** | **0.753** | **0.906**
> ___
> **Q2. Ablation for what layers of $G_{0}$ be used for fusion.**
>
> Following your suggestion, we additionally provide both quantitative and qualitative ablation results for the inversion performance with fusion in
> different layers (see Appendix A.3.4 of the revised manuscript for qualitative results). Note that in our main experiment, we apply feature fusion in layers $\ell_{f}=7$ and $9$, and wavelet fusion in layer $\ell_{w}=11$. Each layer corresponds to fusion of spatial features with resolution $64 \times 64$ and $128 \times 128$, and wavelet coefficients of dimension $w \in \mathbb{R}^{12\times 128 \times 128}.$
>
> From the table below, we observed that the feature fusion on two layers $\ell_{f}=7$ and $9$ showed better reconstruction accuracy than on a single layer $\ell_{f}=7$. Additionally, wavelet fusion in lower layers ($\ell_{w}<11$) was not sufficient enough to preserve the high-fidelity details, especially in the high-frequency region i.e., $L_{wave}$. Wavelet fusion in the higher layer ($\ell_{w}=13$) also degraded the inversion performance, which can be more carefully observed in Figure 10. Overall, our scenario empirically showed
> the most promising reconstruction quality, generating realistic images with the least distortion.
> Feature fusion | Wavelet fusion | $L_2 \downarrow$ | $L_{wave}\downarrow$ | LPIPS $\downarrow$ | SSIM $\uparrow$ |ID sim$\uparrow$
> -------|-------|--------|-------|--------|--------|--------
> $\ell_f=7$  | $\ell_w=7$  |0.028 | 0.359 | 0.365 | 0.667 | 0.796
> |  | $\ell_w=9$  | 0.026 | 0.356 | 0.362 | 0.701 | 0.830
> |  | $\ell_w=11$   | 0.026 | 0.325 | 0.364 | 0.727 | 0.847
> |  | $\ell_w=13$  | 0.024 | 0.314 | 0.366 | 0.727 | 0.845
> $\ell_f=7, 9$ | $\ell_w=7$  | 0.020 | 0.327 | 0.346 | 0.711 | 0.849
> |  | $\ell_w=9$  | 0.016 | 0.289 | 0.330 | 0.724 | 0.880
> |  |$\ell_w=11$  | **0.011** | **0.230** | **0.277** | **0.753** | **0.906**
> |  |$\ell_w=13$  | 0.020 | 0.307 | 0.342 | 0.722 | 0.861
> ___
>
> **Q3. Provided empirical results are not sufficiently convincing for showing the frequency bias of $L_{2}$.**
>
> In Theorem 1, we proved that $L_{2}$ is the sum of $L_{2,f}$ with the same weights of $f \in $ {$LL, HL, LH, HH$}, which is fair theoretically.
> However, as shown in Figure 2(b) and the table below, the majority of $L_{2}$ is originated from $L_{2,LL}$.
> In the light of this observation, we argue that optimizing with $L_{2}$ inevitably induces the bias on $L_{2,LL}$, i.e., low-frequency bias.
> Consequently, we propose wavelet loss, which explicitly measures the discrepancy between generated and ground truth images of high-frequency sub-bands.
>
>  Model | $L_2$ | $L_{2,LL}$ | $L_{2,LH}$ | $L_{2,HL}$ | $L_{2,HH}$
> -------|-------|--------|-------|--------|-------
> $\text{Restyle}_{e4e}$  | 0.0493 | 0.0462 | 0.0010 | 0.0014 | 0.0003
> HyperStyle           | 0.0271 | 0.0245 | 0.0010 | 0.0012 | 0.0002
> HFGI               | 0.0238  | 0.0205 | 0.0009 | 0.0014 | 0.0002
> **WaGI**               | **0.0112**  | **0.0095**  |  **0.0006**  | **0.0008**  | **0.0002**

---

### Official Review · Reviewer_pcW6 · 2022-10-23

**Confidence:** 1
**Clarity, Quality, Novelty And Reproducibility:** This paper is well written. The metho…
**Correctness:** 3
**Technical Novelty And Significance:** 3
**Empirical Novelty And Significance:** 3
**Recommendation:** 6

**Strength And Weaknesses:**

Strength:
1. The performance of this paper is impressive.
2. This paper is well written.

Weakness:
1. In table 4, the performance of WaGI is 0.011(L2). In table 6, why the best performance is only 0.015 for model already adding Wavelet fusion?
2. Artifact problem.  In Figure 6, we can see that even though the proposed method obtain sharper edge, there exists noticeable artifact in the edge part. The artifact looks like ghosting artifact around edgy. Such phenomenon can be also find in Figure 5. I think this artifact is caused by the proposed method. I suggest authors add more discussion about this part.

**Summary Of The Paper:**

This paper find that the normally used L2 loss is biased on low-frequency by using the wavelet transform. Thus this paper proposed the wavelet-based GAN inversion model to effectively lowers distortions on both the low-frequency and high-frequency sub-bands.

**Summary Of The Review:**

My major considersion is the ghosting artifact caused by the proposed method.

---

> ### Author Response · Authors · 2022-11-15
> **Response to Reviewer pcW6**
>
> We are sincerely thankful to every reviewer for reading our research narrowly and giving us thoughtful feedback.
> We are glad to give the impression to all reviewers that our method is reasonable and effective.
> We carefully respond to each of the concerns and questions, together with a revised manuscript that reflects all comments.
> ___
> **Q1. $L_{2}$ values in Table 1 and Table 2 are different.**
>
> Thank you for the thoughtful comment. We assume the tables you refer to are Table 1 and 2 instead of Table 4 and 6. In Table 2, we calculated metrics with images that are tough to invert, e.g., containing complex backgrounds or obstacle occlusion, among CelebA-HQ test images. In the revised manuscripts, we updated the values by evaluation with the total CelebA-HQ test images.
> ___
> **Q2. Artifacts occur around the edge.**
>
> Thanks for your careful attention.
> In Figure 6, an overlapped obstacle becomes marginally transparent, which occurs as an artifact, as you mentioned.
> This phenomenon only occurs in a few cases, in that we cannot find similar artifacts in other figures, such as Figure 4 or 5.
> Moreover, only our model can reconstruct the exact shape of obstacles among all the state-of-the-art GAN inversion models [Yuval et al., 2022; Wang et al., 2022].
> ___
> - [Yuval et al., 2022] Hyperstyle: Stylegan Inversion With Hypernetworks for Real Image Editing, CVPR 2022.
> - [Wang et al., 2022] High-fidelity GAN Inversion for Image Attribute Editing, CVPR 2022.

---

### Official Review · Reviewer_ruHv · 2022-10-24

**Confidence:** 4
**Correctness:** 3
**Technical Novelty And Significance:** 3
**Empirical Novelty And Significance:** 3
**Recommendation:** 5

**Clarity, Quality, Novelty And Reproducibility:**

Clarity, Quality and Reproducibility of this work is ok.
Novelty is not strong enough.

**Strength And Weaknesses:**

Strength:
The overall idea is quite clear and easy to follow, the inversion results are impressive.

Weaknesses:

1. The main concern is the generalization ability of this method for different GAN models. The designed module of Wavelet loss and Wavelet fusion rely on 'SWAGAN: A Style-based WAvelet-driven Generative Model'. I cannot see how the method can be applied to general GAN models, which will severely constrain the application of this method. This constraint also makes the contribution & impact of this work to be weak.

2. Some findings claimed in this work is not new, and actually well-known and explored, e.g., image L2 loss is biased to low-frequency, similar claim is also delivered in 'A Wavelet-Based Generation Network for Image Inpainting'.

**Summary Of The Paper:**

This work aims to solve the inherent limitations in both structural and training aspects in the existing GAN inversion, which preclude the reconstruction of high-frequency features. They show that L2 loss in GAN inversion is biased to reconstruct low-frequency features. To overcome this problem, they introduce WaGI which enables to handle high-frequency features by using a wavelet-based loss term and a wavelet fusion scheme.

**Summary Of The Review:**

As mentioned above, the generalization ability of this method is the main concern. Some findings are also not new.

---

> ### Author Response · Authors · 2022-11-15
> **Response to Reviewer ruHv**
>
> We are sincerely thankful to every reviewer for reading our research narrowly and giving us thoughtful feedback.
> We are glad to give the impression to all reviewers that our method is reasonable and effective.
> We carefully respond to each of the concerns and questions, together with a revised manuscript that reflects all comments.
> ___
> **Q1. Generalization ability of the proposed method to different GAN models.**
>
> As you pointed out, we humbly agree that our method is only applicable to SWAGAN at the current state. As this may be seen as a major constraint of our work, we believe that other generators can also be extended into the wavelet domain with the potential of SWAGAN's performance. Yet, SWAGAN is currently the only wavelet-based generator able to create realistic high-fidelity images and at the same time, capable of GAN inversion and disentangled attribute editing.
>
> We would also like to clarify that we employed SWAGAN as our backbone generator due to the incorporation of wavelets throughout its layers, which enforces the model to be aware of the high-frequency content. Our wavelet fusion leverages this property by the explicit manipulation of wavelet coefficients from the intermediate layers of SWAGAN. As SWAGAN is an extension of StyleGAN2 with progressive growth in the wavelet domain, we also look forward to the expansion of other generators in a similar manner. We would like to leave the appliance of our method for high-frequency aware GAN inversion with various generators for future work.
> ___
> **Q2. Proposed findings are already well-known and explored.**
>
> As you mentioned, many previous works pointed out the bias itself of existing loss terms. However, our contribution can be distinguished in two ways.
> First, we propose a novel loss term that enables the explicit handling of high-frequency features.
> The paper [Yu et al., 2021] you mentioned handles the high-frequency features, but it is different from ours in that it restores them with extra layers, not with a unified loss term.
> Moreover, we propose a novel wavelet-based loss term that enables high-frequency details to be preserved without modifying the network structure.
> Second, we theoretically prove the frequency bias of $L_{1}$ and $L_{2}$ with the observation.
> To the best of our knowledge, we made the first approach to quantitatively prove the frequency bias in the aspect of the wavelet transform.
> ___
> - [Yu et al., 2021] WaveFill: A Wavelet-based Generation Network for Image Inpainting, ICCV 2021.

---

> > ### Comment · Reviewer_ruHv · 2022-12-12
> > **Thanks for your response**
> >
> > Thank you for taking the time to clarify my questions. I have read the author's response and also other reviews. My main concern is still the generalization performance of this work. The SWAGAN backbone is really a significant limitation of this work. It is very a harsh condition to require the wavelet expansion of other generators. I incline to keep my original rating.

---

> > > ### Author Response · Authors · 2022-12-12
> > > **Response to reviewer ruHv's concern.**
> > >
> > > Dear reviewer ruHV,
> > >
> > > We deeply appreciate your thoughtful reply to our response. We also notice the double facet of our work, where leveraging SWAGAN can be a major strength and, at the same time, a weakness. As the extension of a generator to the wavelet domain was introduced most recently, we still exploited this generator to venture the ideation of wavelet-based GAN inversion. Thus, we are willing to see our work as a pioneer among high-frequency aware GAN inversion methods emerging in the near future.
> > >
> > > Yours sincerely,
> > > Authors.

---

### Official Review · Reviewer_FJXP · 2022-10-24

**Confidence:** 3
**Correctness:** 3
**Technical Novelty And Significance:** 3
**Empirical Novelty And Significance:** 3
**Recommendation:** 6

**Clarity, Quality, Novelty And Reproducibility:**

This paper is well-written. The overall quality is good, and the method is somewhat new. The reproducibility is ensured by the code that well be released after the review.

**Strength And Weaknesses:**

Strength:
1. This paper notices an important problem of the existing GAN inversion models that L2 loss is biased to reconstruct low-frequency features.
2. The motivation of each module in the proposed WaGI is reasonable and well explained.

Weakness:
1. The Theorem 1 proves that the L2 loss seems a fair loss without frequency bias, but then the authors also claim that they empirically find the existence of this frequency bias. What is the possible reason that causes the controversy between the theorem and the empirically findings?
2. The authors claim that the wavelet fusion module can transfer high-frequency knowledge without degradation due to the hierarchical upsampling structure of SWAGAN. Does this mean that the wavelet fusion module is specialized to SWAGAN as generator, and the wavelet fusion module cannot be adapted to the generator without a hierarchical upsampling structure? I am really curious about the performance of using other models as generator.
3. It's better to provide figures or some simple experimental results as evidences to support the claim in page 5 that "a substantial amount of image details are placed below f_{nyq/2}".
4. It seems like the ground truth images in Figures 5 and 10 are actually the input images. Please check these figures again.


**Summary Of The Paper:**

This paper focuses on the problem that existing GAN inversion models fail to maintain high-frequency features precisely. In this paper, the authors prove that the widely used L2 loss term is biased towards the low-frequency components. To solve this problem, the authors attempt to interpret GAN inversion in the frequency domain and propose a method, WaGI, that contains a new wavelet loss and a wavelet fusion module. The proposed method is proved to be the SOTA GAN inversion model.

**Summary Of The Review:**

Generally, this paper notices an important obstacle of the existing methods for the GAN inversion task and proposes a method to solve this problem. However, conditioned upon the issues mentioned in [Weakness], I would give a score 6 for now.

---

> ### Author Response · Authors · 2022-11-15
> **Response to Reviewer FJXP**
>
> We are sincerely thankful to every reviewer for reading our research narrowly and giving us thoughtful feedback.
> We are glad to give the impression to all reviewers that our method is reasonable and effective.
> We carefully respond to each of the concerns and questions, together with a revised manuscript that reflects all comments.
> ___
> **Q1. Controversy between the theorem and the empirical findings.**
>
> First, thanks for the meaningful comment.
> In Theorem 1, we proved that $L_{2}$ is a sum of $L_{2, f}$, which is not biased on specific $f \in $ { $LL, LH, HL, HH$ }.
> However, as shown in Figure 2(b) and the table below, the value of $L_{2,LL}$ is significantly larger than the others.
> Consequently, merely using loss terms obtained by adding all $L_{2,f}$ with the same weights induces the bias on $L_{2,LL}$, i.e., low-frequency bias.
>  Model | $L_2$ | $L_{2,LL}$ | $L_{2,LH}$ | $L_{2,HL}$ | $L_{2,HH}$
> -------|-------|--------|-------|--------|-------
> $\text{Restyle}_{e4e}$  | 0.0493 | 0.0462 | 0.0010 | 0.0014 | 0.0003
> HyperStyle           | 0.0271 | 0.0245 | 0.0010 | 0.0012 | 0.0002
> HFGI               | 0.0238  | 0.0205 | 0.0009 | 0.0014 | 0.0002
> **WaGI**               | **0.0112**  | **0.0095**  |  **0.0006**  | **0.0008**  | **0.0002**
> ___
> **Q2. Generalization of proposed methods to other generators without a hierarchical upsampling structure.**
>
> We would first like to clarify that we employed SWAGAN as our backbone generator due to the incorporation of wavelets throughout its generator, which enforces the model to be aware of the high-frequency content. Our wavelet fusion leverages this property by the explicit manipulation of wavelet coefficients from intermediate layers of SWAGAN.
> As SWAGAN is an extension of StyleGAN2 with progressive growth in the wavelet domain, we believe that any generator extended with wavelet reconstruction alike can be compatible with our proposed methods. Yet, SWAGAN is the only wavelet-based generator able to create realistic high-fidelity images and at the same time, capable of GAN inversion and disentangled attribute editing. We are also curious how our method will work for other generators extended in the wavelet domain and would like to leave it as future work.
> ___
> **Q3. Evidences to show the amount of information placed below $\frac{f_{nyq}}{2}$.**
>
> In Appendix A.2.2 of the revised manuscript figure, we provide images that omit the high-frequency wavelet coefficients at $1^{st}$, $2^{nd}$, and $3^{rd}$ wavelet decomposition, respectively.
> In Figure 9, we can check that the image omitting only the $1^{st}$ wavelet coefficients (Config A), i.e., omitting information between $f_{nyq}/2$ and $f_{nyq}$, still preserves most of the image details.
> In contrast, omitting information between $f_{nyq}/2^2$ and $f_{nyq}/2$, or $f_{nyq}/2^3$ and $f_{nyq}/2^2$ severely degrades the image details.
> In other words, the majority of visible image details are placed below $f_{nyq}/2$.
> This result justifies the design of multi-level wavelet loss, which covers broader frequency ranges than a single-level wavelet loss.
> ___
> **Q4. Images in Figure 5 and 10 are actually the input images.**
>
> Again, thanks for the precise comment. Ground truth for the editing results do not exist. We fixed the caption in the revised manuscript.

---

### Author Response · Authors · 2022-11-15
**General Response**

Dear reviewers and AC,

We fully appreciate your effort to review our manuscript and provide meaningful feedback thoroughly.

As reviewers summarized, our work solves the inherent problem of GAN inversion methods failing to preserve the high-frequency features (FJXP, ruHV) via applying wavelet loss and wavelet fusion (FJXP, ruHv, pcW6, Xb3x) with a wavelet-based generator (Xb3x). Our method is efficient enough to minimize distortions of all frequency ranges and is currently the SOTA GAN inversion model.

In response to the comments, we have revised the draft with the following additional discussions and experiments:
- Ablation study of the choice of fusion layers (in Appendix A.3.4)
- Ablation study of the design of the fusion method (in Appendix A.3.4)
- Updated experimental results with fair evaluation dataset (Table 2)
- Additional discussions on the information gap between frequency sub-bands (in Appendix A.2.2)
- Minor correction of caption (Figure 5, 12)

These updates are temporarily written in “$\color{blue}{\text{blue}}$
” for your convenience in the revised manuscript.
We believe your comments helped our manuscript become more clear and more constructive.

Thank you very much!

Authors.

---

### Decision · Program_Chairs · 2023-01-20

**Decision:**

Reject

**Justification For Why Not Higher Score:**

I think that the method is limited in application to a very narrow setup, and that the claimed theoretical contribution is a trivial, well known observation (to which the reviewers agree). Both is not reflected in the paper, and therefore the paper is not ready for publication in my opinion.



**Justification For Why Not Lower Score:**

N/A

**Metareview: Summary, Strengths And Weaknesses:**

The paper argues that existing approaches to inverting GANs are inherently limited at reconstructing high-frequency information, and proposes an method for the inversion of GAN models that aims at lowering distortions. The proposed method builds on the recent HFGI (Wang et al., 2022a) and the SWAGAN (Gal et al., 2021) papers, and enables inversion for the architecture used in the SWAGAN paper. The method is shown to perform well for this architecture.

There are two major weaknesses:

First, the applicability of the approach for different GAN models beyond the SWAGAN model is unclear. The method pertains to SWAGAN. In the rebuttal, the authors confirm that the method is only applicable to SWAGAN at the current state, and would like to leave the adoption/application to other generators to future work. However, this feature limits the applicability of the proposed approach, and is therefore important to address in the current paper. In addition, this feature is not sufficiently reflected in the abstract and introduction, so from reading the abstract a reader would think that the method is more generally applicable.

Second, the paper notes that it proves that the l2 loss is biased to low frequencies, and lists this as a major contribution (i.e., this is listed first in the contribution section). What the paper provides in terms of a proof is a simple statement saying that all frequencies in the l2 loss are weighted the same, which is trivial, and concludes that since images have most of it's energy on low-frequencies (also well known), the l2 loss is biased to low-frequencies.


**Summary Of Ac-Reviewer Meeting:**

Here is a summary of the reviews and how they were weighted.

R1 notes that the paper finds an important obstacle for inverting GANs and proposes a solution for it. Notes that 'each module is reasonable and well explained'.
In terms of weaknesses, the reviewer identifies a controversy between theory and empirical results, specifically, theorem 1 proves that the l2 loss seems to be good and does not have a frequency bias, but empirically the authors find a frequency bias. The authors note that the values of the loss corresponding to low-frequency components is relatively large compared to the others; however this is not surprising since images are known to be biased toward low-frequency signals, so this is not a satisfying answer.

R2 notes that 'The overall idea is quite clear and easy to follow, the inversion results are impressive.'
Two weaknesses are identified, which are the major weaknesses of the paper in my opinion:

The generalization ability for different GAN models is unclear, it to SWAGAN, which limits the applicability of the work. The authors confirm that the method is only applicable to SWAGAN at the current state, and would like to leave the adoption/application to other generators to future work.

The reviewer also notes that the image l2 loss being biased to low-frequencies is well known. The authors respond that while this is indeed known, they make the first approach to provide a proof. I agree with the reviewer that this is well known, does not require proof, and does not constitute a significant contribution.

R3, with low confidence, notes that the performance is impressive and that the paper is well written. The reviewer notes that while the method obtains sharper edges, this comes at the cost of increased artifacts. The authors respond that this only occurs in some places, but do not quantify this further.
I did not take this review into account, since the reviewer said that they are unable to assess the manuscript.

R4 is the most positive review. The reviewer writes: 'The paper is a natural extension of the HFGI and SWAGAN papers. The authors addressed critical points, in the continuations of the ideas presented in the mentioned papers to extended the HFGI method. In qualitative evaluation the method show better results on the inversion task, especially on text, which is usually a high frequency signal. In addition, it seems like this method is better than HFGI for editing tasks (visually).'

I agree, but the reviewer also notes:

'The paper is an incremental work combining ideas from two previous papers, namely, SWAGAN and HFGI, yet the implementation is not trivial and the results look good.'

'The authors write in the abstract that they proved the loss used for others models is biased to the low frequency features. First, the analysis show the loss is unbiased. Second, I'm not sure if the empirical evidence they provide in order to show the loss is indeed biased in practice, is sufficiently convincing.'

Thus, the reviewer also acknowledges the two major weaknesses noted in the meta review.

I think that the method is limited in application to a very narrow setup, and that the claimed theoretical contribution is a relatively trivial, well-known observation (to which the reviewers agree). Both is not reflected in the paper, and therefore the paper is not ready for publication in my opinion.